# Proxy Target: Bridging the Gap Between Discrete Spiking Neural Networks and Continuous Control

**Zijie Xu**
Peking University
Beijing, China 100871
zjxu25@stu.pku.edu.cn

**Tong Bu**
Peking University
Beijing, China 100871
putong30@pku.edu.cn

**Zecheng Hao**
Peking University
Beijing, China 100871
haozecheng@pku.edu.cn

**Jianhao Ding**
Peking University
Beijing, China 100871
djh01998@alumni.pku.edu.cn

**Zhaofei Yu**[*]
Peking University
Beijing, China 100871
yuzf12@pku.edu.cn

## Abstract

Spiking Neural Networks (SNNs) offer low-latency and energy-efficient decision making on neuromorphic hardware, making them attractive for Reinforcement Learning (RL) in resource-constrained edge devices. However, most RL algorithms for continuous control are designed for Artificial Neural Networks (ANNs), particularly the target network soft update mechanism, which conflicts with the discrete and non-differentiable dynamics of spiking neurons. We show that this mismatch destabilizes SNN training and degrades performance. To bridge the gap between discrete SNNs and continuous-control algorithms, we propose a novel proxy target framework. The proxy network introduces continuous and differentiable dynamics that enable smooth target updates, stabilizing the learning process. Since the proxy operates only during training, the deployed SNN remains fully energy-efficient with no additional inference overhead. Extensive experiments on continuous control benchmarks demonstrate that our framework consistently improves stability and achieves up to 32% higher performance across various spiking neuron models. Notably, to the best of our knowledge, this is the first approach that enables SNNs with simple Leaky Integrate and Fire (LIF) neurons to surpass their ANN counterparts in continuous control. This work highlights the importance of SNN-tailored RL algorithms and paves the way for neuromorphic agents that combine high performance with low power consumption. Code is available at https://github.com/xuzijie32/Proxy-Target.

## 1 Introduction

Reinforcement Learning (RL), combined with Artificial Neural Networks (ANNs), has become a cornerstone of modern artificial intelligence, achieving remarkable success in diverse domains such as game playing [Mnih, 2013, Silver et al., 2016, Mnih et al., 2015], autonomous driving [Kiran et al., 2021, Sallab et al., 2017, Shalev-Shwartz et al., 2016], and large language model training [Ouyang et al., 2022, Bai et al., 2022, Shao et al., 2024]. Among these, continuous control problems have drawn particular attention due to their close alignment with real-world robotic and embodied AI applications [Kober et al., 2013, Gu et al., 2017, Brunke et al., 2022]. However, the high computational cost and power demands of ANN-based RL algorithms limit their deployment on edge devices such as drones, wearables, and IoT sensors [Abadía et al., 2021, Tang et al., 2020, Yamazaki et al., 2022].

---

[*]Corresponding author

39th Conference on Neural Information Processing Systems (NeurIPS 2025).

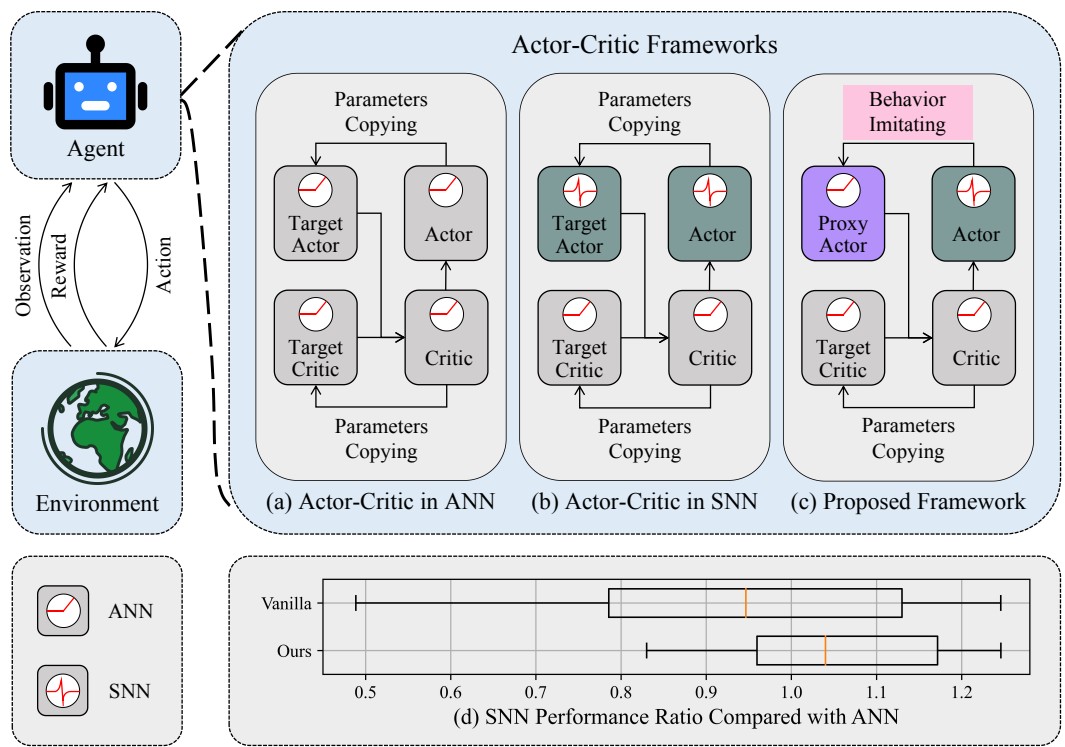

Figure 1: Overview of the training framework and performance comparison. (a)-(c) are different training paradigms. (a) Actor-Critic framework in ANNs, (b) The Actor-Critic framework for SNNs, (c) the proposed proxy target framework for SNNs. (d) Performance ratio of SNNs relative to ANNs across five random seeds and five environments. The middle orange line denotes the median, the box spans from the first to the third quartile, and the whiskers extend to the farthest data within 1.5 inter-quartile range from the box.

Inspired by biological neural systems, Spiking Neural Networks (SNNs) offer sparse, event-driven computation with ultra-low latency and energy consumption on neuromorphic hardware [DeBole et al., 2019, Davies et al., 2018]. These properties make SNNs attractive for RL applications on resource-constrained edge devices [Yamazaki et al., 2022]. Recent works have attempted to integrate SNNs into continuous-control RL algorithms via hybrid frameworks [Tang et al., 2020, 2021, Zhang et al., 2024, Chen et al., 2024, Zhang et al., 2022], where a spiking actor network (SAN) is co-trained with an ANN critic using Spatio-Temporal Backpropagation (STBP) [Wu et al., 2018, Fang et al., 2021], as illustrated in Fig. 1(b). With well-chosen hyperparameters, such frameworks have shown that SNNs can approach or even surpass the performance of ANNs in some tasks.

However, most of these studies simply retrofit SNNs into existing ANN-centric RL frameworks without adapting the algorithms to SNN dynamics. Since ANNs and SNNs exhibit fundamentally different computational characteristics, it remains unclear **whether RL algorithms designed for continuous, differentiable activations are well-suited for discrete, event-driven networks**.

A key issue arises from the *target network soft update mechanism*, a core component widely used in off-policy RL algorithms to stabilize training by gradually updating target networks [Sutton and Barto, 2018, Lillicrap, 2015, Fujimoto et al., 2018, Haarnoja et al., 2017]. This mechanism relies on continuous, smooth output changes—a property violated by the non-differentiable, binary nature of SNN spikes. This can cause abrupt output shifts, leading to unstable optimization objective, and oscillatory updates. Such instability not only makes the model highly sensitive to random seed initialization but also hampers convergence and undermines reliability in real-world deployment.

To address this mismatch between discrete spikes and continuous-control updates, we propose a proxy target framework for SNN-based RL (Fig. 1(c)). Instead of using an SNN target actor, we introduce a differentiable proxy actor network that imitates the behavior of the online spiking actor network.

The proxy target network can alter its output smoothly and continuously, stabilizing the learning process and improving performance, as demonstrated in Fig. 1(d). Since the proxy target network is only used for auxiliary training, the proposed approach retains SNN's advantages of low-latency and energy efficiency during inference in real world applications. Our main contributions are summarized as follows:

- We identify a critical mismatch between discrete SNN outputs and the continuous target network soft update mechanism used in off-policy RL, showing how this conflict destabilizes training and degrades performance.

- We propose a proxy target framework that replaces the spiking target network with a continuous, differentiable proxy, enabling smooth target updates and stable optimization.

- We introduce an implicit gradient-based update rule that aligns the proxy with the online SNN, mitigating target output gaps and giving precise optimization goals.

- Extensive experiments across multiple neuron models and continuous control benchmarks demonstrate consistent stability improvements and up to **32**% higher average performance. To the best of our knowledge, this is the first approach where SNNs with simple Leaky Integrate-and-Fire (LIF) neurons surpass ANN performance in continuous control.

## 2 Related works

### 2.1 Learning rules of SNN-based RL

**Synaptic plasticity.** Inspired by the plasticity of biological synapses, several works have integrated SNNs into reinforcement learning via reward-modulated spike-timing-dependent plasticity (R-STDP) [Florian, 2007, Frémaux and Gerstner, 2016, Gerstner et al., 2018, Frémaux et al., 2013, Yang et al., 2024]. These approaches are biologically plausible and energy-efficient, but have limited performance on complex tasks.

**ANN-SNN conversion.** With the progress of ANN-based deep RL and ANN-SNN conversion algorithms [Cao et al., 2015, Bu et al., 2022a,b], some studies [Patel et al., 2019, Tan et al., 2021, Kumar et al., 2025] convert well-trained Deep Q-Networks (DQNs) [Mnih, 2013, Mnih et al., 2015] into SNNs. Such conversion-based methods achieve lower energy consumption during inference, but require ANN pre-training.

**Gradient-based direct training.** To avoid ANN pre-training, several works [Liu et al., 2022, Chen et al., 2022, Qin et al., 2022, Sun et al., 2022, 2025] directly train SNNs for RL using STBP [Wu et al., 2018], while Bellec et al. [2020] introduced e-prop with eligibility traces to learn policies through the policy gradient algorithm [Sutton et al., 1999]. These approaches achieve competitive results in discrete action spaces, but they cannot be extended to continuous control tasks.

### 2.2 Hybrid framework of spiking actor network.

In continuous-control problems where the action space is continuous, hybrid frameworks have been extensively explored. Tang et al. [2020] first proposed an SNN-based actor co-trained with an ANN critic in the Actor-Critic framework [Konda and Tsitsiklis, 1999]. Tang et al. [2021] demonstrated that population encoding improves the performance of spiking actor networks. Subsequent works enhanced these frameworks through various mechanisms, such as utilizing dynamic neurons [Zhang et al., 2022], incorporating lateral connections [Chen et al., 2024], adding bio-plausible topologies [Zhang et al., 2024], and integrating dynamic thresholds [Ding et al., 2022].

While these hybrid approaches report performance comparable to or exceeding their ANN counterparts, two key limitations remain. First, they often rely on complex neuron models (e.g., current-based LIF or second-order dynamic neurons), increasing computational cost and training difficulty. Second, the RL algorithms themselves are not modified to account for SNN-specific dynamics, which may cause instability and suboptimal convergence. In contrast, our proxy target framework is tailored to the discrete, event-driven nature of SNNs, achieving superior stability and performance with only simple LIF neurons.

## 3 Preliminaries

To avoid ambiguity, we use *training steps* to denote RL time steps and *simulation steps* to denote internal SNN simulation time steps.

### 3.1 Reinforcement Learning

Reinforcement Learning (RL) involves an agent interacting with an environment. The agent observes the current state $s$, performs an action $a$ and receives a reward $r$, while the environment transitions to the next state $s'$. The agent's objective is to learn a policy $\pi_\phi$, parameterized by $\phi$, that maximizes the expected return.

In continuous control settings, the action space is a continuous vector (e.g., torque values). Most continuous control algorithms adopt the Actor–Critic framework with a deterministic policy [Sutton and Barto, 2018], where the actor $\pi_\phi$ outputs actions $a = \pi_\phi(s)$ and the critic $Q_\theta$ evaluates them with parameters $\theta$ [Konda and Tsitsiklis, 1999]. The actor is updated by the deterministic policy gradient [Silver et al., 2014]:

$$\nabla_\phi J(\phi) = \mathbb{E}\left[ \nabla_a Q_\theta(s,a) \mid_{a=\pi(s)} \nabla_\phi \pi_\phi(s) \right]. \tag{1}$$

The critic is updated via temporal-difference (TD) learning [Sutton, 1988] using the Bellman equation [Bellman, 1966]:

$$Q_\theta(s,a) \leftarrow y, \quad y = r + \gamma Q_{\theta'}(s',a'), a' = \pi_{\phi'}(s'), \tag{2}$$

where $\gamma$ is the discount factor and $(\pi_{\phi'}, Q_{\theta'})$ denote target networks.

### 3.2 Target network soft update

The target networks $(\pi_{\phi'}, Q_{\theta'})$ share the same architecture as their online counterparts $(\pi_\phi, Q_\theta)$ but are updated more slowly to provide stable learning targets. Their parameters are updated by the Polyak function with smoothing factor $\tau$:

$$\phi' \leftarrow \tau\phi + (1-\tau)\phi', \quad \theta' \leftarrow \tau\theta + (1-\tau)\theta'. \tag{3}$$

These soft updates play a crucial role in off-policy continuous-control algorithms. As shown in Eqs. (1, 2), the actor and critic are jointly optimized through bootstrapping, which can cause oscillatory updates due to their mutual dependence. The target networks mitigate this by producing slowly changing targets, thereby stabilizing training and preventing divergence.

### 3.3 Spiking Neural Networks

**Spiking neuron model.** In an SNN, each neuron integrates presynaptic spikes into its membrane potential and emits a spike when the potential exceeds a threshold. The Leaky Integrate and Fire (LIF) neuron [Gerstner and Kistler, 2002] is one of the most widely used models, governed by the following dynamics:

$$I_t^l = W^l S_t^{l-1} + b^l, \quad H_t^l = \lambda V_{t-1}^l + I_t^l, \tag{4}$$

$$S_t^l = \Theta(H_t^l - V_{th}), \quad V_t^l = (1 - S_t^l)H_t^l + S_t^l \cdot V_{\text{reset}}, \tag{5}$$

where $I$ is the input current, $H$ is the accumulated membrane potential, $S$ is the binary output spike, $V$ is the membrane potential after the firing process. $W$ and $b$ are the weights and the biases, $V_{th}$, $V_{\text{reset}}$, and $\lambda$ are the threshold voltage, the reset voltage and the membrane leakage parameter, respectively. All subscripts $(\cdot)_t$ and all superscripts $(\cdot)^l$ denote simulation step $t$ and layer $l$ respectively. $\Theta(\cdot)$ is the Heaviside function.

**Spiking actor network.** The spiking actor network (SAN) consists of a population encoder with Gaussian receptive fields [Tang et al., 2021], a multi-layer SNN, and a decoder that uses the membrane potentials of non-firing neurons as continuous outputs [Chen et al., 2024]. The SAN is trained using STBP with a surrogate gradient function. Detailed forward and backward formulations are provided in Appendix A.2.

# 4   Methodology

In this section, we propose a novel proxy target framework to address the incompatibility between the discrete dynamics of spiking neurons and the continuous target network soft update mechanism in RL. Section 4.1 analyzes the instability caused by discrete target outputs and introduces a proxy target network with continuous dynamics. Section 4.2 presents an implicit imitation mechanism that aligns the proxy network with the online SNN through gradient-based optimization. Section 4.3 summarizes the overall training procedure.

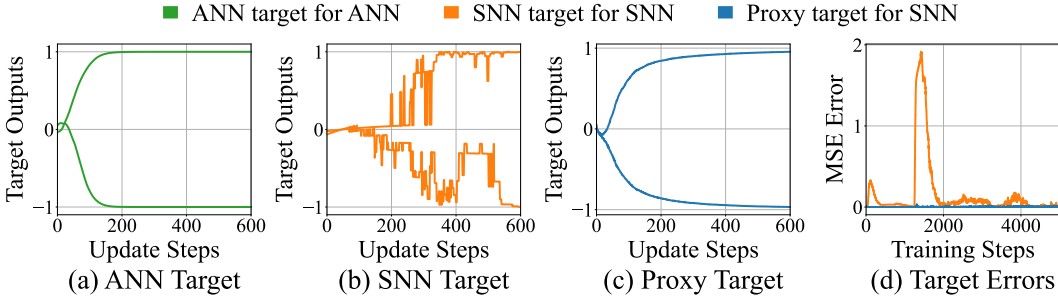

Figure 2: Effects of different target network update mechanisms. (a)-(c) show output trajectories of different target networks during updates, where each line denotes a normalized output dimension within $(-1, 1)$. (a) ANN target network exhibits smooth transitions; (b) SNN target network produces discrete and irregular output jumps; (c) the proposed proxy target achieves continuous and stable transitions. (d) Mean squared error between target and online networks during training in the InvertedDoublePendulum-v4 environment.

## 4.1   Addressing discrete targets by proxy network

**Performance degradation due to discrete target outputs.**   In the standard Actor–Critic framework, the target network is updated using the Polyak function (Eq. 3), which assumes that small parameter updates lead to smooth output transitions. This assumption holds for ANNs with continuous activation functions but fails for SNNs, whose firing function is binary and non-differentiable. To illustrate this effect, we construct target networks corresponding to trained online networks using identical architectures and neuron models. The target parameters are updated according to Eq. 3 with $\tau = 0.005$ (the most commonly used setting), while the online network is frozen. Figures 2(a)–(b) show the target outputs during updates: the ANN target evolves smoothly, whereas the SNN target exhibits frequent discontinuous jumps. Although both targets eventually converge to their online counterparts, the discrete shifts in the SNN target (Fig. 2(b)) cause erratic transitions that propagate instability to the critic's optimization objectives, resulting in oscillatory and unreliable learning dynamics [Fujimoto et al., 2018].

**Smoothing target outputs by proxy network.**   As illustrated in Fig. 1(c), to restore smoothness, we introduce a proxy target network that replaces the discrete spiking neurons of SNNs with continuous activation functions of ANNs. As shown in Fig. 2(c), the proxy network produces gradual output transitions during updates, effectively eliminating the discrete jumps observed in SNN targets. This design enables stable soft updates and prevents abrupt shifts in the target outputs, thereby improving the stability of the overall Actor–Critic learning process.

## 4.2   Addressing target output gaps by implicit updates

**Performance degradation due to target output gaps.**   Although the proxy network achieves smooth updates, directly substituting spiking neurons with continuous activations (e.g., ReLU) introduces an output gap between the proxy and the online SNN. This discrepancy prevents the proxy target from accurately reproducing the output of the online SNN, distorting the critic's learning targets and reducing overall policy performance.

**Aligning proxy network by implicit updates.**   Since the approximation errors cannot be eliminated by explicitly copying the weights of the online SNN, we propose an implicit proxy update method.

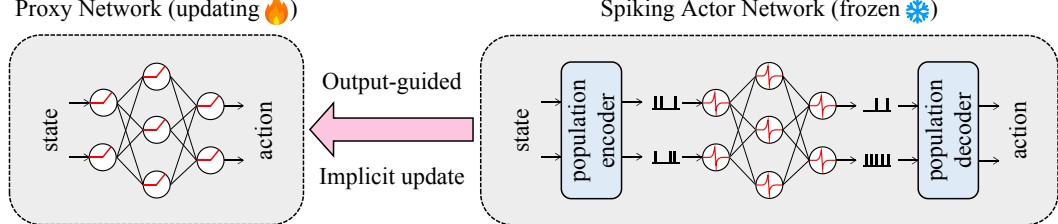

Figure 3: Architecture of the proposed proxy network and the spiking actor network. The proxy actor is updated implicitly by imitating the behavior of the online spiking actor network, ensuring stable and accurate target updates.

As shown in Fig. 3, unlike the explicit soft update that directly averages parameters, our approach computes updates in the output space, which gradually reduces the gap between the online SNN and the proxy target. Let the proxy actor $\pi_{\phi'}^{\text{Proxy}}$ have parameters $\phi'$ and the online spiking actor $\pi_{\phi}^{\text{SNN}}$ have parameters $\phi$. For each input state $s$, the proxy output is implicitly updated toward the SNN actor output as:

$$\pi_{\phi'}^{\text{Proxy}}(s) \leftarrow (1 - \tau') \cdot \pi_{\phi'}^{\text{Proxy}}(s) + \tau' \cdot \pi_{\phi}^{\text{SNN}}(s). \tag{6}$$

where $\tau'$ is a smoothing coefficient similar to $\tau$ in Eq. 3. Since it is difficult to directly update the corresponding parameter according to Eq. 6, we instead perform a gradient-based optimization that achieves a similar effect:

$$\phi' \leftarrow \phi' + \tau \left( \pi_{\phi}^{\text{SNN}}(s) - \pi_{\phi'}^{\text{Proxy}}(s) \right) \nabla_{\phi'} \pi_{\phi'}^{\text{Proxy}}(s) = \phi' - \frac{\tau}{2} \nabla_{\phi'} \left\| \pi_{\phi'}^{\text{Proxy}}(s)) - \pi_{\phi}^{\text{SNN}}(s)) \right\|_2^2, \tag{7}$$

where $\| \cdot \|_2^2$ denotes the squared $\ell_2$ norm. Thus, the proxy network can be updated by gradient descent that minimizes the proxy loss:

$$L_{proxy} = \frac{1}{N} \sum_{i=1}^{N} \left\| \pi_{\phi'}^{\text{Proxy}}(s_i)) - \pi_{\phi}^{\text{SNN}}(s_i)) \right\|_2^2, \tag{8}$$

where $N$ denotes the batch size, $s_i$ are the states sampled from the replay buffer in RL algorithm. This proxy update mechanism acts as a form of implicit imitation learning, aligning the proxy network with the SNN actor while maintaining smooth output transitions, as demonstrated in Theorem 1.

**Theorem 1** *Let the proxy network $\pi_{\phi'}^{Proxy}$ be updated by minimizing the loss $L_{proxy}$ in Eq. 8. During each update, as the proxy learning rate $lr_{proxy} \to 0$, the output change satisfies*

$$\| \pi_{\phi'_{new}}^{Proxy}(s) - \pi_{\phi'_{old}}^{Proxy}(s) \| \to 0,$$

*where $\phi'_{old}$ and $\phi'_{new}$ denote parameters before and after the update, respectively. Hence, minimizing $L_{proxy}$ ensures sufficiently small and smooth policy updates, promoting stable optimization.*

Since the proxy network is a multi-layer feedforward model, a universal approximator [Hornik et al., 1989], it can asymptotically match the SNN actor's output by minimizing Eq. 8. To further demonstrate this empirically, Fig. 2(d) shows the mean-squared output gap between the proxy and the SNN actor during training. While the SNN target occasionally diverges from the online SNN, the proxy network remains well-aligned throughout, validating that the proposed approach effectively mitigates target output gaps and provides precise and stable optimization goals for RL training.

### 4.3 Overall training framework

The proposed proxy target framework is shown in Fig. 1(c). The proxy actor network contains continuous activations of ANN that replace the discontinuous SNN target actor network. Instead of explicitly updating network parameters, the proxy actor is implicitly optimized to imitate the behavior of the online SNN actor by minimizing the loss in Eq. 8. During each update episode, the proxy actor is optimized for $K$ iterations to reliably approximate the discrete SNN outputs, compensating for the greater representational difficulty. Meanwhile, the target critic is updated explicitly by copying

**Algorithm 1** Proxy Target framework

---

1: Initialize SNN actor network $\pi_\phi^{\text{SNN}}(s)$, ANN critic network $Q_\theta^{\text{ANN}}(s,a)$ with parameters $\phi, \theta$
2: Initialize proxy actor $\pi_{\phi'}^{\text{Proxy}}(s)$ and ANN target critic $Q_{\theta'}^{\text{ANN}}(s,a)$ with parameters $\phi'$ and $\theta'$
3: Initialize replay buffer $\mathcal{D}$
4: **for** each iteration **do**
5:     Execute action $a$ according to $\pi_\phi^{\text{SNN}}(s)$ and store the transition $(s,a,r,s')$ in $\mathcal{D}$
6:     **if** proxy target update **then**
7:         **for** $k = 1$ to $K$ **do**
8:             Sample a minibatch of $N$ transitions $(s_i, a_i, r_i, s_i')$ from $\mathcal{D}$
9:             Update proxy actor parameters $\phi'$ by minimizing:

$$L_{proxy} = \frac{1}{N} \sum_i \left\| \pi_{\phi'}^{\text{Proxy}}(s_i) - \pi_\phi^{\text{SNN}}(s_i) \right\|_2^2$$

10:         **end for**
11:     **end if**
12:     **if** ANN target update **then**
13:         Update ANN target critic parameters $\theta'$ by the Polyak function: $\theta' \leftarrow \tau\theta + (1-\tau)\theta'$
14:     **end if**
15:     **if** ANN critic update **then**
16:         Compute target values $y_i$ using proxy actor $\pi_{\phi'}^{\text{Proxy}}$ and target critic $Q_{\theta'}^{\text{ANN}}$
17:         Update ANN critic by minimizing: $L_{critic} = \frac{1}{N}\sum_i \left(Q_\theta^{\text{ANN}}(s_i, a_i) - y_i\right)^2$
18:     **end if**
19:     **if** SNN actor update **then**
20:         Update SNN actor by maximizing: $J = \frac{1}{N}\sum_i Q_\theta^{\text{ANN}}\left(s_i, \pi_\phi^{\text{SNN}}(s_i)\right)$
21:     **end if**
22: **end for**

---

weights using the Polyak function, as both the critic and target critic are conventional ANNs. The complete training procedure is summarized in Algorithm 1.

As demonstrated in Theorem 1 and Fig. 2(c)-(d), the proxy actor not only produces smooth output transitions but also closely tracks the SNN actor's behavior[1]. The proxy target framework effectively alleviates the instability caused by discrete and imprecise targets in traditional SNN–RL frameworks, resulting in a more stable training process within the Actor–Critic framework.

It is worth noting that the proposed mechanism preserves the energy efficiency of SNNs, as the proxy network and the critic network are used exclusively during training and are discarded during deployment, introducing no additional computational overhead during deployment.

## 5 Experiments

### 5.1 Experimental setup

The proposed proxy target framework (PT) was evaluated across multiple continuous-control tasks in the MuJoCo simulator [Todorov et al., 2012, Todorov, 2014b] using the OpenAI Gymnasium benchmark suite [Brockman, 2016, Towers et al., 2024], including `InvertedDoublePendulum-v4` (IDP) [Todorov, 2014a], `Ant-v4` [Schulman et al., 2015], `HalfCheetah-v4` [Wawrzyński, 2009], `Hopper-v4` [Erez et al., 2012], and `Walker2d-v4`. All environments follow the default configurations without modifications.

The experiments were carried out with different spiking neuron models, such as the LIF neuron, the current-based LIF neuron (CLIF) Tang et al. [2021], and the dynamic neuron (DN) Zhang et al.

---

[1]Minor fluctuations in the proxy network output (Fig. 2(c)) resemble the stochasticity introduced by soft target updates with noise injection in DRL [Fujimoto et al., 2018], which can further reduce overfitting in value estimation.

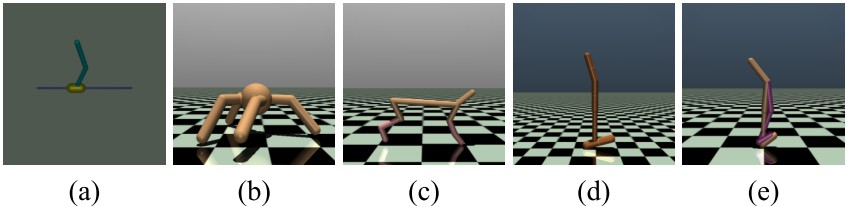

(a)       (b)       (c)       (d)       (e)

Figure 4: Continuous control tasks of the MuJoCo environments on OpenAI Gymnasium. (a) InvertedDoublePendulum-v4, (b) Ant-v4, (c) HalfCheetah-v4, (d) Hopper-v4, (e) Walker2d-v4.

[2022]. The LIF and CLIF neuron parameters follow Tang et al. [2021], while the DN parameters are initialized as in Zhang et al. [2022].

We tested the proposed algorithm in conjunction with the TD3 algorithm [Fujimoto et al., 2018], all detailed parameter settings are provided in Appendix A.4. For a fair comparison, all spiking actor networks share the same architecture, encoding, and decoding schemes provided in Appendix A.2. All SNNs have a simulation step of **5** unless otherwise noted. All reported data in this section are reproduced results across five random seeds.

## 5.2   Results across different spiking neurons

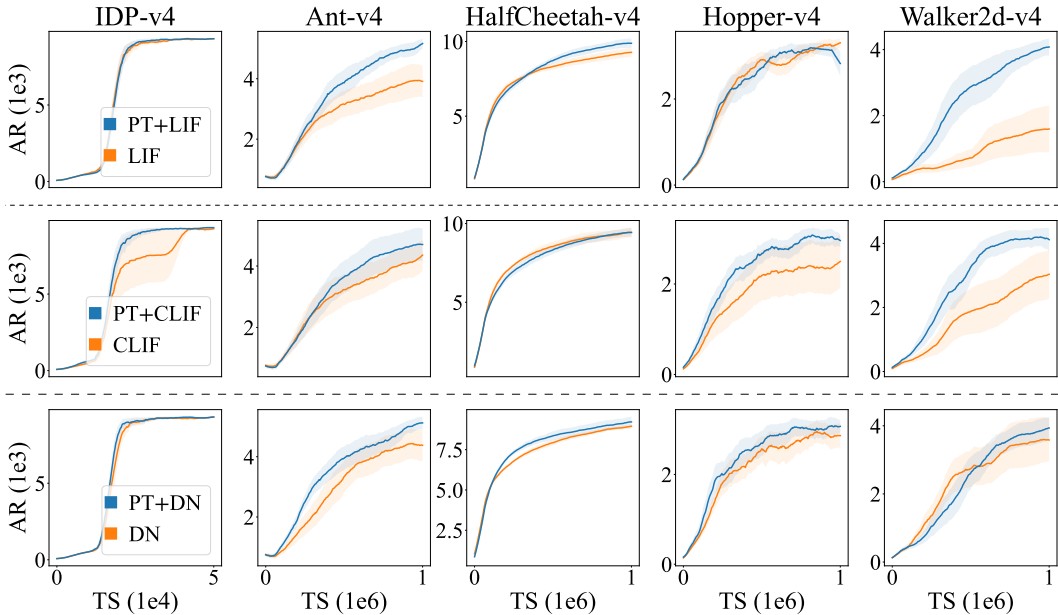

Figure 5: Learning curves of the proxy target framework (PT) and the vanilla Actor–Critic framework with the LIF neuron, the CLIF neuron and the DN. AR denotes average returns, and TS denotes training steps. The shaded region represents half a standard deviation over 5 different seeds. Curves are uniformly smoothed for visual clarity.

**Increasing performance.**   Fig. 5 shows the learning curves of the proposed proxy target framework and the vanilla Actor-Critic framework with different spiking neurons. The proxy target framework improves the performance of different spiking neurons, demonstrating its general applicability in delivering both faster convergence and higher final returns across different neuron types and environments.

**Improving stability.**   Fig. 6(a) shows the performance variance (after training) of the proxy target framework and the vanilla Actor-Critic framework with different spiking neurons. The proxy target framework reduces the variance of different spiking neurons, demonstrating its capability to stabilize

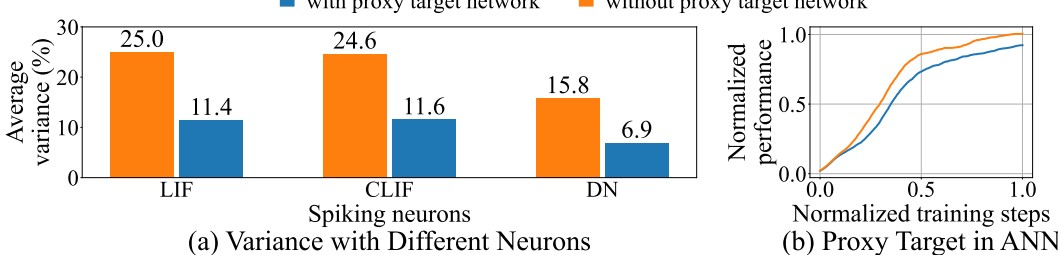

Figure 6: (a) Average variance of different neurons after training. The average variance is computed by averaging the standard deviation ratio with 5 seeds, across all environments. (b) Normalized learning curves across all environments of the ANN integrated with the proposed proxy network across all environments. The performance and training steps are normalized linearly to $(0, 1)$. Curves are uniformly smoothed for visual clarity.

training. This is crucial for real-world deployments, where retraining costs are high and consistent behavior is required.

## 5.3 Exceeding state-of-the-art

To quantify relative improvements, we define the average performance gain (APG) as:

$$APG = \left( \frac{1}{|\text{envs}|} \sum_{\text{env} \in \text{envs}} \frac{\text{performance}(\text{env})}{\text{baseline}(\text{env})} - 1 \right) \cdot 100\%, \tag{9}$$

where $|\text{envs}|$ denotes the total number of environments, performance(env) and baseline(env) are the performance of the algorithm and the baseline in that particular environment. Tab.1 compares our proxy target framework with ANN-based RL, the ANN–SNN conversion method [Bu et al., 2025] (100 simulation steps), and other state-of-the-art SNN-based RL algorithms, including pop-SAN [Tang et al., 2021], MDC-SAN [Zhang et al., 2022], and ILC-SAN [Chen et al., 2024]. With the proxy network, a simple LIF-based SNN surpasses all baselines, including those using complex neuron dynamics or connection structures, and achieves higher average returns than standard ANNs. Although performance varies across tasks, the average gain across all environments and neurons indicates the general applicability of the proxy target framework.

Table 1: Max average returns over 5 random seeds with different spiking neurons, and the average performance gain against the ANN baseline, where $\pm$ denotes one standard deviation.

| Method | IDP-v4 | Ant-v4 | HalfCheetah-v4 | Hopper-v4 | Walker2d-v4 | APG |
|---|---|---|---|---|---|---|
| ANN (TD3) | $7503 \pm 3713$ | $4770 \pm 1014$ | $10857 \pm 475$ | $3410 \pm 164$ | $4340 \pm 383$ | $0.00\%$ |
| ANN-SNN | $3859 \pm 4440$ | $3550 \pm 963$ | $8703 \pm 658$ | $3098 \pm 281$ | $4235 \pm 354$ | $-21.11\%$ |
| Vanilla LIF | $9347 \pm 1$ | $4294 \pm 1170$ | $9404 \pm 625$ | $3520 \pm 94$ | $1862 \pm 1450$ | $-10.54\%$ |
| pop-SAN | $9351 \pm 1$ | $4590 \pm 1006$ | $9594 \pm 689$ | $2772 \pm 1263$ | $3307 \pm 1514$ | $-6.66\%$ |
| MDC-SAN | $9350 \pm 1$ | $4800 \pm 994$ | $9147 \pm 231$ | $3446 \pm 131$ | $3964 \pm 1353$ | $0.37\%$ |
| ILC-SAN | $9352 \pm 1$ | $5584 \pm 272$ | $9222 \pm 615$ | $3403 \pm 148$ | $4200 \pm 717$ | $4.64\%$ |
| PT-CLIF | $9351 \pm 1$ | $5014 \pm 1074$ | $9663 \pm 426$ | $3526 \pm 112$ | $4564 \pm 555$ | $5.46\%$ |
| PT-DN | $9350 \pm 1$ | $5400 \pm 277$ | $9347 \pm 666$ | $3507 \pm 144$ | $4277 \pm 650$ | $5.06\%$ |
| PT-LIF | $9348 \pm 1$ | $5383 \pm 250$ | $10103 \pm 607$ | $3385 \pm 157$ | $4314 \pm 423$ | $\mathbf{5.84\%}$ |

## 5.4 Simple neurons perform best

Interestingly, the simplest LIF neuron achieves the highest overall performance under the proxy target framework. This contrasts with previous findings where complex neuron models generally perform better. Once the SNN surpasses its ANN counterpart, the primary performance bottleneck shifts from the neuron model to the RL algorithm itself. Hence, introducing more complex spiking dynamics may unnecessarily increase training difficulty and even degrade performance.

## 5.5 SNN-friendly design

Fig. 6(b) shows the normalized performance of ANN with and without the proxy network. The proxy target framework cannot improve the performance in ANNs, confirming that the observed benefits arise from addressing SNN-specific challenges rather than providing a stronger RL algorithm. This validates the SNN-friendly design of our framework.

## 5.6 Energy efficiency

Finally, we evaluate inference energy consumption across models. The comparison includes a traditional ANN-based TD3 model, a baseline spiking actor network using vanilla LIF neurons, and our PT-LIF model. The consumption is estimated as the same way as Merolla et al. [2014], where multiply-accumulate (MAC) operation costs 3.97pJ on modern NPUs[2] [Millar et al., 2025] and synaptic operation (SOP) costs 77fJ [Hu et al., 2021].

Table 2: Energy consumptions of different tasks per inference for the spiking actor network with LIF neurons, where the energy unit is nano-joule (nJ).

| Method | IDP-v4 | Ant-v4 | HalfCheetah-v4 | Hopper-v4 | Walker2d-v4 | Average |
|---|---|---|---|---|---|---|
| ANN (TD3) | 72.33 | 295.60 | 283.41 | 274.26 | 283.41 | 281.78 (71014.4 MACs) |
| Vanilla LIF | 8.14 | 11.78 | 15.13 | 7.21 | 18.82 | 12.21 ($158.6 \times 10^3$ SOPs) |
| PT-LIF | 9.01 | 12.18 | 13.46 | 6.86 | 13.93 | **11.09** ($144.0 \times 10^3$ SOPs) |

As shown in Tab.2, the ANN (TD3) model consumes significantly more energy, while both spiking models demonstrate dramatically lower energy consumption. Specifically, our proposed PT-LIF model achieves the lowest average consumption while maintaining better stability and performance. Moreover, PT-LIF's average firing rate (32%) is slightly lower than that of the vanilla LIF model (33%), further improving energy efficiency. These results highlight the superior energy efficiency of the proposed method, making it compelling for deployment on energy-constrained platforms.

## 6 Conclusion

In this work, we identified a critical mismatch between the discrete dynamics of SNNs and the continuous requirement of the target network soft update mechanism in the Actor-Critic framework. To address this, we proposed a novel proxy target framework that enables smooth target updates and faster convergence. Experimental results demonstrate that the proxy network can stabilize training and improve performance, enabling simple LIF neurons to surpass ANN performance in continuous control.

In contrast to previous works which retrofit SNNs into ANN-centric RL frameworks, this work opens a door to investigate and design SNN-friendly RL algorithms which is tailored to SNN's specific dynamics. In the future, more SNN-specific adjustments could be applied to SNN-based RL algorithms to improve performance and energy efficiency in real-world, resource-constrained RL applications.

**Limitation.** While this work designs a proxy target framework that is suitable for SNN-based RL, it still remains at the simulation level. The next step may involve implementing it on edge devices and enabling decisions-making in the real world.

## 7 Acknowledgement

This work is funded by National Natural Science Foundation of China (62422601, U24B20140, and 62088102), Beijing Municipal Science and Technology Program (Z241100004224004) and Beijing Nova Program (20230484362, 20240484703), and National Key Laboratory for Multimedia Information Processing.

---

[2]Energy per MAC is estimated from the maximum inference throughput ($mJ^{-1}$) across seven NPUs: MAX78k (C/R), GAP8, NXP-MCXN947, HX-WE2 (S/P), and MILK-V [Millar et al., 2025]. NPU initialization energy is excluded.

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

# A Appendix

## A.1 Proof of Theorem 1

**Theorem 1** *Let the proxy network $\pi_{\phi'}^{Proxy}$ be updated by minimizing the loss $L_{proxy}$ in Eq. 8. During each update, as the proxy learning rate $lr_{proxy} \to 0$, the output change satisfies*

$$\|\pi_{\phi'_{new}}^{Proxy}(s) - \pi_{\phi'_{old}}^{Proxy}(s)\| \to 0,$$

*where $\phi'_{old}$ and $\phi'_{new}$ denote parameters before and after the update, respectively. Hence, minimizing $L_{proxy}$ ensures sufficiently small and smooth policy updates, promoting stable optimization.*

**Proof 1** *By standard gradient descent, we have:*

$$\lim_{lr_{proxy} \to 0} \|(\phi'_{new} - \phi'_{old})\| = \lim_{lr_{proxy} \to 0} \left\| lr_{proxy} \cdot \nabla_{\phi'_{old}} L_{proxy} \right\| = 0.$$

*Under the assumption that $\pi_{\phi'}^{Proxy}(s)$ is continuously differentiable with respect to $\phi'$, we apply a first-order Taylor expansion:*

$$\lim_{lr_{proxy} \to 0} \left\| \pi_{\phi'_{new}}^{Proxy}(s) - \pi_{\phi'_{old}}^{Proxy}(s) \right\| = \lim_{lr_{proxy} \to 0} \left\| (\phi'_{new} - \phi'_{old}) \nabla_{\phi'} \pi_{\phi'_{old}}^{Proxy}(s) \right\| = 0.$$

## A.2 Spiking actor network architecture

The spiking actor network (SAN) consists of a population encoder with Gaussian receptive fields, a multi-layer SNN with population output, and a decoder with non-firing neurons.

### A.2.1 Forward propagation of the SAN

In the state encoder, each input dimension consists of $N_{in}$ soft reset IF neurons with different Gaussian receptive fields with trainable parameters $\mu$ and $\sigma$. The neurons receive a stimulation $A_E$ at every time step and outputs spikes $S^{in}$ according to:

$$A_E = \exp\left[ -\frac{1}{2} \frac{(s - \mu)^2}{\sigma^2} \right], \tag{10}$$

$$
\begin{aligned}
V_t^{in} &= V_{t-1}^{in} - S_{t-1}^{in} + A_E, \\
S_t^{in} &= \Theta(V_t^{in} - V_E),
\end{aligned} \tag{11}
$$

where $V_E$ is the threshold of the encoding populations.

The last layer of the SNN consists of $N_{out}$ neurons for each action dimension, respectively. The decoder layer is made up of non-spiking integrate neurons connected to the last layer of SNN:

$$V_t^{out} = V_{t-1}^{out} + W^{out} \cdot S_t^L + b^{out}, \tag{12}$$

where $W^{out}$ and $b^{out}$ are weights and biases. The final output action is determined by the membrane potential in the last time step $a = V_T^{out}$. The detailed forward propagation of the spiking actor network is shown in Algo. 2.

### A.2.2 Back propagation of the SAN

SAN parameters are trained by the gradient with respect to the output action $\frac{\partial L}{\partial a}$, where $a = V_T^{out}$.

The output decoder can be updated by:

$$
\begin{aligned}
\frac{\partial L}{\partial W^{out}} &= \frac{\partial L}{\partial a} \cdot \frac{\partial V_T^{out}}{\partial W^{out}} \\
\frac{\partial L}{\partial b^{out}} &= \frac{\partial L}{\partial a} \cdot \frac{\partial V_T^{out}}{\partial b^{out}}
\end{aligned} \tag{13}
$$

Then, the main SNN is trained by STBP with the rectangular surrogate function defined as:

$$\Theta'(x) = \begin{cases} \frac{1}{2\omega}, & -\omega \le x \le \omega \\ 0, & \text{else} \end{cases}, \tag{14}$$

---

**Algorithm 2** Forward propagation of spiking actor network

---

1: **Input:** $M_s$-dimensional observation $s$
2: Compute the stimulation of input populations:

$$A_E = \exp\left[-\frac{1}{2}\frac{(s-\mu)^2}{\sigma^2}\right]$$

3: **for** t=1,…,T **do**
4:    Compute the output spikes of the population encoder:

$$V_t^{in} = V_{t-1}^{in} - S_{t-1}^{in} + A_E$$

$$S_t^{in} = \Theta(V_t^{in} - V_E)$$

5:    **for** l=1,…,L **do**
6:       Update neurons in layer $l$ at timestep $t$
7:    **end for**
8:    Update the decoder neurons' membrane potential:

$$V_t^{out} = V_{t-1}^{out} + W^{out} \cdot S_t^L + b^{out}$$

9: **end for**
10: **Output:** $M_a$-dimensional action $a = V_T^{out}$

---

where $\omega$ is the window size.

Next, the gradient of the encoder stimulation $A_E$ is written in Eq.15. Note that $\frac{\partial S_t^{in}}{\partial A_E}$ is manually set to 1 to simplify the gradient computation.

$$\frac{\partial L}{\partial A_E} = \sum_{t=1}^{T} \frac{\partial L}{\partial S_t^{in}} \cdot \frac{\partial S_t^{in}}{\partial A_E} = \sum_{t=1}^{T} \frac{\partial L}{\partial S_t^{in}} \tag{15}$$

Finally, the trainable parameters $\mu$ and $\sigma$ in the encoder can be updated by:

$$\begin{array}{c}\frac{\partial L}{\partial \mu} = \frac{\partial L}{\partial A_E} \cdot \frac{\partial A_E}{\partial \mu} = \frac{\partial L}{\partial A_E} \cdot \frac{s-\mu}{\sigma^2} A_E \\ \frac{\partial L}{\partial \sigma} = \frac{\partial L}{\partial A_E} \cdot \frac{\partial A_E}{\partial \sigma} = \frac{\partial L}{\partial A_E} \cdot \frac{(s-\mu)^2}{\sigma^3} A_E\end{array} \tag{16}$$

### A.3    Other Spiking Neuron Models

Section 3.3 already shows the LIF neuron model, this section will show two other spiking neuron models conducted in the experiments.

#### A.3.1    Current-Based LIF neuron model

In the current-based LIF (CLIF) neurons proposed in Tang et al. [2021], the input current in Eq.4 is redefined as:

$$I_t^l = \alpha I_{t-1}^l + W^l S_t^{l-1} + b^l, \tag{17}$$

where $\alpha$ is the current leakage parameter. While other dynamics of CLIF neurons are the same as those of LIF neurons.

#### A.3.2    Dynamic neuron model

Zhang et al. [2022] designed a second-order dynamic neurons (DN) for continuous control. The DN consists of a membrane potential $V$ and a resistance item $U$ to simulate hyperpolarization. Its dynamics is shown as follows:

$$\frac{dV_t^l}{dt} = V_t^{l\,2} - V_t^l - U_t^l + I_t^l \tag{18}$$

$$\frac{dU_t^l}{dt} = \theta_v V_t^l - \theta_u U_t^l \tag{19}$$

where $\theta_v$ and $\theta_U$ are the conductivities of $V$ and $U$, respectively. Once firing a spike, the membrane potential $V$ is reset to $V_{\text{reset}}$ and the resistance $U$ is added by $\theta_s$.

With a firs-order Taylor expansion, the iterative DN can be written as:

$$
\begin{aligned}
C_t^l &= \alpha \cdot C_{t-1}^l + W^l S_t^{l-1} + b^l; \\
V_t^l &= \left(1 - S_{t-1}^l\right) \cdot V_{t-1}^l + S_{t-1}^l \cdot V_{\text{reset}}; \\
U_t^l &= U_{t-1}^l + S_{t-1}^l \cdot \theta_u; \\
V_{\text{delta}} &= V_t^{l^2} - V_t^l - U_t^l + C_t^l; \\
U_{\text{delta}} &= \theta_v \cdot V_t^l - \theta_u \cdot U_t^l; \\
V_t^l &= V_t^l + V_{\text{delta}}; \\
U_t^l &= U_t^l + U_{\text{delta}}; \\
S_t^l &= \Theta\left(V_t^l - V_{th}\right).
\end{aligned}
\tag{20}
$$

## A.4 Experiment details

### A.4.1 Compute Resources

We conduct the experiments on an RTX 3090 GPU and an Intel(R) Xeon(R) Platinum 8362 CPU.

### A.4.2 Spiking Neuron Parameters

The LIF and CLIF neuron parameters are shown in Tab.3, which are the same as those in Tang et al. [2021], except that the LIF neuron has no current leakage parameter. The DN parameters are shown in Tab.4, determined by the pre-learning process proposed in Zhang et al. [2022].

Table 3: Parameters of LIF and CLIF [Tang et al., 2021] neurons

| Parameter | LIF | CLIF [Tang et al., 2021] |
|---|---|---|
| Membrane leakage parameter $\lambda$ | 0.75 | 0.75 |
| Threshold voltage $V_{th}$ | 0.5 | 0.5 |
| Reset voltage $V_{\text{reset}}$ | 0 | 0 |
| Current leakage parameter $\alpha$ | - | 0.5 |

Table 4: Parameters of the DN [Zhang et al., 2022]

| Parameter | Value |
|---|---|
| SNN time steps | 5 |
| Threshold voltage $V_{th}$ | 0.5 |
| Current leakage parameter $\alpha$ | 0.5 |
| Conductivity of membrane potential $\theta_v$ | $-0.172$ |
| Conductivity of hidden state $\theta_u$ | 0.529 |
| Reset voltage $V_{\text{reset}}$ | 0.021 |
| spike effect to hidden state $\theta_s$ | 0.132 |

### A.4.3 Specific Parameters for the Proxy Target Framework

Tab.5 shows hyper-parameters of the proxy target framework for different spiking neurons. To capture the behavior of the SNN, the hidden sizes of the proxy network is set wider than that of its online SNN. Since different spiking neurons exhibit different dynamics and learning speed, hidden sizes[3] and learning rate of the proxy network vary across spiking neurons. All other hyper-parameters are kept the same.

---

[3]Since the InvertedDoublePendulum environment is relatively easier, there is no needs for such a wide proxy network. Thus we set the hidden size is $(512, 512)$ specifically for that environment for the CLIF neuron.

Table 5: Hyper-parameters of the proxy network framework with different spiking neurons

| Parameter | LIF | CLIF [Tang et al., 2021] | DN [Zhang et al., 2022] |
|---|---|---|---|
| Proxy network architecture | $(512, 512)$ | $(800, 600)$ | $(512, 512)$ |
| Proxy network activation | ReLU | ReLU | ReLU |
| Proxy network learning rate | $1 \cdot 10^{-3}$ | $3 \cdot 10^{-3}$ | $3 \cdot 10^{-3}$ |
| Proxy network optimizer | Adam | Adam | Adam |
| Proxy update iterations $K$ | 3 | 3 | 3 |
| Proxy update batch size $N$ | 256 | 256 | 256 |

### A.4.4 Spiking Actor Network Parameters

All hyper-parameters of the spiking actor network are shown in Tab.6. This is the same as in a wide range of previous studies [Tang et al., 2021, Zhang et al., 2022, Chen et al., 2024].

Table 6: Hyper-parameters of the spiking actor network

| Parameter | Value |
|---|---|
| Encoder population per dimension $N_{in}$ | 10 |
| Encoder threshold $V_E$ | 0.999 |
| Network hidden units | $(256, 256)$ |
| Decoder population per dimension $N_{out}$ | 10 |
| Surrogate gradient window size $\omega$ | 0.5 |

### A.4.5 RL algorithm parameters

We conduct the experiment based on the TD3 algorithm [Fujimoto et al., 2018], with hyper-parameters shown in Tab.7.

Table 7: Hyper-parameters of the implemented TD3 algorithm [Fujimoto et al., 2018]

| Parameter | Value |
|---|---|
| Actor learning rate | $3 \cdot 10^{-4}$ |
| Actor regularization | None |
| Critic learning rate | $3 \cdot 10^{-4}$ |
| Critic regularization | None |
| Critic architecture | $(256, 256)$ |
| Critic activation | ReLU |
| Optimizer | Adam |
| Target update rate $\tau$ | $5 \cdot 10^{-3}$ |
| Batch size $N$ | 256 |
| Discount factor $\gamma$ | 0.99 |
| Iterations per time step | 1.0 |
| Reward scaling | 1.0 |
| Gradient clipping | None |
| Replay buffer size | $10^6$ |
| Exploration noise $\mathcal{N}(0, \sigma)$ | $\mathcal{N}(0, 0.1)$ |
| Actor update interval $d$ | 2 |
| Target policy noise $\mathcal{N}(0, \tilde{\sigma})$ | $\mathcal{N}(0, 0.2)$ |
| Target policy noise clip $c$ | 0.5 |

### A.4.6 Experiment environments

Fig. 4 shows various MuJoCo environments [Todorov et al., 2012, Todorov, 2014b] on OpenAI Gymnasium benchmarks [Brockman, 2016, Towers et al., 2024], including InvertedDoublePendulum (IDP) [Todorov, 2014a], Ant [Schulman et al., 2015], HalfCheetah [Wawrzyński, 2009], Hopper

[Erez et al., 2012] and Walker2d. All environment setups used the default configurations without modifications.

It is worth noting that the state vector ranges from $-\infty$ to $\infty$, it is normalized to $(-1, 1)$ by a tanh function. In addition, since the action has the minimum and maximum limits, the output of actor network is normalized to $(-1, 1)$ by a tanh function and then linearly scaled to $(\text{Min action}, \text{Max action})$.

## A.5 Pseudo codes for the proposed proxy target framework in conjunction with TD3

We present the detailed pseudocode of the general proxy target framework in Algo.1, in Section 4.3. Specifically, Algo.3 shows how to implement the proxy target framework in the TD3 algorithm [Fujimoto et al., 2018]. It is worth noting that the original TD3 algorithm updates the target actor with delay. However, in our framework, the proxy actor is updated without delay because of its inherently slow update pace.

---

**Algorithm 3** Proxy target framework with TD3

---

1: Initialize SNN actor network $\pi_\phi^{\text{SNN}}(s)$, ANN critic networks $Q_{\theta_1}^{\text{ANN}}(s, a)$, $Q_{\theta_2}^{\text{ANN}}(s, a)$ with weights $\phi$, $\theta_1$ and $\theta_2$
2: Initialize proxy actor $\pi_{\phi'}^{\text{Proxy}}(s)$ and ANN target critics $Q_{\theta_1'}^{\text{ANN}}(s, a)$, $Q_{\theta_2'}^{\text{ANN}}(s, a)$ with weights $\phi'$, $\theta_1'$ and $\theta_2'$
3: Initialize replay buffer $\mathcal{D}$
4: **for** each iteration **do**
5:     Execute action $a = \pi_\phi^{\text{SNN}}(s) + \epsilon$, $\epsilon \sim \mathcal{N}(0, \sigma)$ and observe reward $r$ and next state $s'$
6:     Store the transition $(s, a, r, s')$ in $\mathcal{D}$
7:     **for** $k = 1$ to $K$ **do**
8:         Sample a minibatch of $N$ transitions $(s_i, a_i, r_i, s_i')$ from $\mathcal{D}$
9:         Update proxy actor network parameters $\phi'$ by minimizing the loss:

$$L_{target} = \frac{1}{N} \sum_i \left\| \pi_{\phi'}^{\text{Proxy}}(s_i) - \pi_\phi^{\text{SNN}}(s_i) \right\|_2^2$$

10:     **end for**
11:     $y_i = r_i + \gamma \min_{j=1,2} Q_{\theta_j'}^{\text{ANN}}(s_i, \tilde{a}_i)$, $\tilde{a}_i = \pi_{\phi'}^{\text{Proxy}}(s_i) + \epsilon$, $\epsilon \sim \text{clip}\left(\mathcal{N}(0, \tilde{\sigma}), -c, c\right)$
12:     Update ANN critics by minimizing the critic loss: $L_{critic} = \frac{1}{N} \sum_i \left( Q_{\theta_j}^{\text{ANN}}(s_i, a_i) - y_i \right)^2$
13:     **if** $t$ mod $d$ **then**
14:         Update SNN actor by maximizing the objective function: $J = \frac{1}{N} \sum_i Q_{\theta_1}^{\text{ANN}}\left(s_i, \pi_\phi^{\text{SNN}}(s_i)\right)$
15:         Update ANN critic target parameters explicitly by the Polyak function: $\theta_j' \leftarrow \tau\theta_j + (1-\tau)\theta_j'$
16:     **end if**
17: **end for**

---

## A.6 Additional experiments results

### A.6.1 Additional results in terms of performance

In the main text, we show that our proxy target framework can increase performance for various spiking neurons. Fig. 7 shows the normalized learning curves of our proxy target framework for different spiking neurons. In addition, Tab. 8, Tab. 9, and Tab .10 show the maximum average returns and the average performance gains of the proxy network against vanilla SNN with LIF neuron, CLIF neuron, and dynamic neuron, respectively.

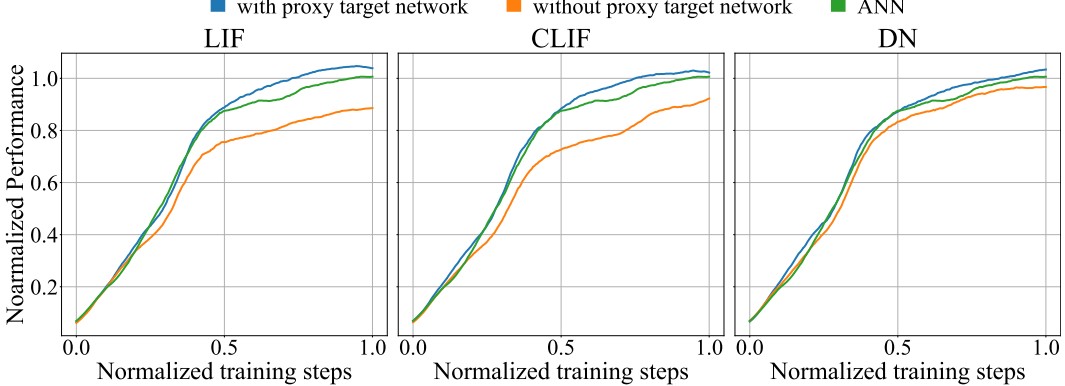

Figure 7: Normalized learning curves of the proposed proxy target framework with different spiking neurons across all environments. The performance and training steps are normalized linearly based on ANN performance. Curves are uniformly smoothed for visual clarity.

Table 8: Max average returns over 5 random seeds with LIF neurons.

| Method | IDP | Ant | HalfCheetah | Hopper | Walker2d | APG |
|---|---|---|---|---|---|---|
| Vanilla LIF | $9347 \pm 1$ | $4294 \pm 1170$ | $9404 \pm 625$ | $\mathbf{3520 \pm 94}$ | $1862 \pm 1450$ | $\mathbf{32.15\%}$ |
| PT-LIF | $\mathbf{9348 \pm 1}$ | $\mathbf{5383 \pm 250}$ | $\mathbf{10103 \pm 607}$ | $3385 \pm 157$ | $\mathbf{4314 \pm 423}$ | |

Table 9: Max average returns over 5 random seeds with CLIF neurons.

| Method | IDP | Ant | HalfCheetah | Hopper | Walker2d | APG |
|---|---|---|---|---|---|---|
| Vanilla CLIF | $\mathbf{9351 \pm 1}$ | $4590 \pm 1006$ | $9594 \pm 689$ | $2772 \pm 1263$ | $3307 \pm 1514$ | $\mathbf{15.03\%}$ |
| PT-CLIF | $9351 \pm 1$ | $\mathbf{5014 \pm 1074}$ | $\mathbf{9663 \pm 426}$ | $\mathbf{3526 \pm 112}$ | $\mathbf{4564 \pm 555}$ | |

Table 10: Max average returns over 5 random seeds with dynamic neurons.

| Method | IDP | Ant | HalfCheetah | Hopper | Walker2d | APG |
|---|---|---|---|---|---|---|
| Vanilla DN | $9350 \pm 1$ | $4800 \pm 994$ | $9147 \pm 231$ | $3446 \pm 131$ | $3964 \pm 1353$ | $\mathbf{4.87\%}$ |
| PT-DN | $9350 \pm 1$ | $\mathbf{5400 \pm 277}$ | $\mathbf{9347 \pm 666}$ | $\mathbf{3507 \pm 144}$ | $\mathbf{4277 \pm 650}$ | |

### A.6.2 Additional results in ANN

We show the normalized learning curves of the proxy target framework with ANN in Fig. 6(b). Here, we show the detailed learning curves and maximum average returns of 5 environments in Fig. 8 and Tab.11, respectively.

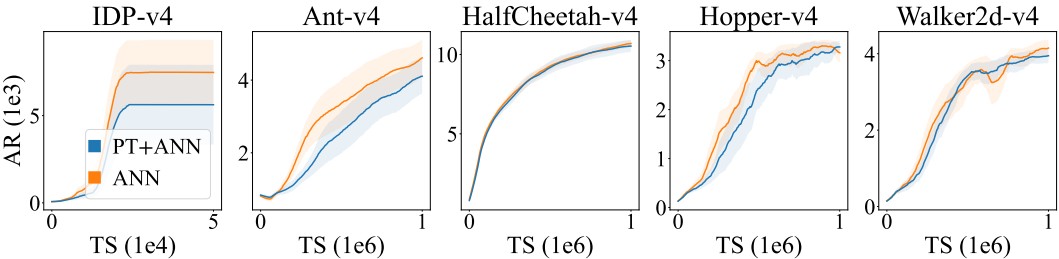

Figure 8: Learning curves of utilizing the proxy target framework in ANN. The PT represents the proxy target framework, AR denotes average returns, and TS is training steps. The shaded region represents half a standard deviation over 5 different seeds. Curves are uniformly smoothed for visual clarity.

Table 11: Max average returns over 5 random seeds with ANN (TD3).

| Method | IDP | Ant | HalfCheetah | Hopper | Walker2d | APG |
|--------|-----|-----|-------------|--------|----------|-----|
| ANN | $\mathbf{7503 \pm 3713}$ | $\mathbf{4770 \pm 1014}$ | $\mathbf{10857 \pm 475}$ | $3410 \pm 164$ | $\mathbf{4340 \pm 383}$ | $-\mathbf{8.38}\%$ |
| PN-ANN | $5653 \pm 4540$ | $4234 \pm 998$ | $10708 \pm 773$ | $\mathbf{3435 \pm 145}$ | $4106 \pm 366$ | |

