# OpenReview forum: "Proxy Target: Bridging the Gap Between Discrete Spiking Neural Networks and Continuous Control"
_NeurIPS.cc/2025/Conference — NeurIPS 2025 poster_

### Official Review · Reviewer_tLiq · 2025-06-21

**Clarity:** 3
**Significance:** 3
**Originality:** 2
**Rating:** 5
**Confidence:** 3

**Summary:**

This paper investigates the challenge of applying Spiking Neural Networks (SNNs) to continuous control reinforcement learning (RL) tasks—a domain where conventional Artificial Neural Networks (ANNs) dominate due to their compatibility with smooth gradient-based updates. The authors point out an incompatibility between the discrete, event-driven nature of SNNs and the soft-update mechanism of target networks in RL, which relies on continuous and differentiable dynamics. To solve this problem, the paper introduces a proxy target framework that decouples the discrete behavior of SNNs from the smooth updating required in RL. This proxy serves as a differentiable surrogate for SNN-based target networks, enabling stable and effective learning. Experimental results across multiple continuous control benchmarks (e.g., MuJoCo tasks) show that the proposed proxy target mechanism can improve the performance of SNN agents—achieving up to 32% performance gains over baseline SNNs—without requiring major changes to the overall learning algorithm or spiking neuron models.

**Questions:**

1. Can the authors provide empirical evidence or simulated estimates to support the claim that SNNs offer superior energy efficiency for edge deployment compared to ANN baselines running on modern NPUs? Specifically, how does the energy consumption of your SNN-based agent compare (in hardware, simulator, or profiling estimates) to that of ANN agents with similar model sizes on commercially available low-power accelerators such as those described in https://arxiv.org/pdf/2503.22567?

**Ethical Concerns:**

["NO or VERY MINOR ethics concerns only"]

**Final Justification:**

I read the rebuttal carefully. The authors answered my questions. So I decide to raise my score.

**Limitations:**

yes

**Quality:**

3

**Strengths And Weaknesses:**

## Strengths

1. **Timely and Relevant Problem**.
The paper tackles the critical gap in making SNNs viable for continuous control RL tasks on edge devices. Given the growing interest in neuromorphic computing for energy-constrained agents (e.g., robotics), this work is both relevant and potentially impactful.

2. **Novel Technical Contribution**.
The proxy target mechanism is a clean and elegant solution to the incompatibility between spike-based updates and target network soft updates. By introducing a continuous and differentiable proxy to interface with the RL loss, the paper bridges a gap that has previously hindered the use of SNNs in continuous action spaces.

3. **Empirical Validation**.
The experiments are extensive and compare different spiking neuron models under standard RL benchmarks. The gains in learning stability and final performance are significant, demonstrating the effectiveness of the proposed technique across neuron types.

## Weaknesses

1. **Energy Efficiency Claims Not Substantiated**.
While the paper argues that SNNs are better suited for edge deployment due to energy efficiency, no empirical or simulated hardware evaluation is provided to validate this claim. In fact, modern NPUs and embedded accelerators (e.g., those discussed in https://arxiv.org/pdf/2503.22567) can run small ANNs at milliwatt-level power budgets—potentially lower than neuromorphic chips like Intel’s Loihi-2 (~1W) (https://open-neuromorphic.org/neuromorphic-computing/hardware/loihi-2-intel/). The authors should temper their claims or support them with energy and latency measurements.

---

> ### Author Rebuttal · Authors · 2025-07-31
>
> We are pleased that you appreciate our proposed concept of the proxy target. Additionally, we are delighted to see the significant progress modern NPUs have made in reducing the energy consumption of ANNs. We would like to address your concerns as follows.
>
> >While the paper argues that SNNs are better suited for edge deployment due to energy efficiency, no empirical or simulated hardware evaluation is provided to validate this claim.
>
> Thank you for this valuable suggestion. Recent studies have demonstrated that SNNs can achieve higher energy efficiency in FPGAs through empirical hardware simulations [1]. We will add this to the Introduction part of the manuscript to support the energy efficiency claim.
>
> >In fact, modern NPUs and embedded accelerators (e.g., those discussed in [2]) can run small ANNs at milliwatt-level power budgets—potentially lower than neuromorphic chips like Intel’s Loihi-2 (~1W)[3].
>
> Intel’s Loihi-2 [2] is not designed to achieve extremely energy efficient, as it has to compromise for compatibility with complex neurons and brain-inspired learning rules. For instance, it supports graded spike events (up to 32 bit) and three-factor learning rules. Thus, we believe it is fairer to compare the energy consumption of ANNs running on NPUs with SNNs running on FPGAs. The detailed comparison will be shown in the answer to the next question.
>
> >Can the authors provide empirical evidence or simulated estimates to support the claim that SNNs offer superior energy efficiency for edge deployment compared to ANN baselines running on modern NPUs? Specifically, how does the energy consumption of your SNN-based agent compare (in hardware, simulator, or profiling estimates) to that of ANN agents with similar model sizes on commercially available low-power accelerators such as those described in [2]?
>
> **ANN inference energy in modern NPUs.** We estimated the energy consumption of each MAC based on the data provided in [2], using the maximum inference per $mJ$ across seven NPUs (i.e. MAX78k (C), MAX78k (R), GAP8, NXP-MCXN947, HX-WE2 (S), HX-WE2 (P),  MILK-V), not including NPU initialization.
>
> **Table R1:** Estimates of energy consumption per MAC in NPU, with data provided in [2].
> ||NAS|ResNet|SimpleNet|Autoenc|YoloV1_small|
> | -------- |:--------: | :--------: | :--------: | :--------: | :--------: |
> |MMACs|74.2512|37.7812|38.0006|0.5455|43.8294|
> |Maximum Inferences time per $mJ$|2.75|9.31|4.13|47.06|5.15|
> |Energy Consumption per MAC ($pJ$)|4.90|4.61|6.37|38.95|3.97|
>
> We then use the minimum estimate (i.e., 3.97 $mJ$ per MAC) to calculate the energy consumption for our baseline ANN model.
>
> **Table R2:** MACs and estimated energy consumption per inference for ANN models running on NPUs in different environments.
>
> |    | IDP-v4   | Ant-v4   |HalfCheetah-v4|Hopper-v4|Walker2d-v4|Average|
> | -------- | :--------: | :--------: | :--------: | :--------: | :--------: | :--------: |
> | MACs  |68608|74496|71424|69120|71424|71014.4|
> |Energy Consumption ($nJ$)|272.33|295.60|283.41|274.26|283.41|281.78|
>
>
> **SNN inference energy estimations.** As shown in the main text, we estimate the energy consumption of SNNs in the same way as [4], where the synaptic operation (SOP) costs $77fJ$ [5,6].
>
>
> **Table R3:** SOPs and estimated energy consumption for vanilla LIF in different environments.
> |    | IDP-v4   | Ant-v4   |HalfCheetah-v4|Hopper-v4|Walker2d-v4|Average|
> | -------- | :--------: | :--------: | :--------: | :--------: | :--------: | :--------: |
> |SOPs (1k)|105.8 |	152.9 |	196.6 	|93.6 	|244.4 |	158.6|
> |Energy Consumptions ($nJ$)|8.14|11.78|15.13|7.21|18.82|12.21|
>
> **Table R4:** SOPs and estimated energy consumption for the proposed PT-LIF in different environments.
> |    | IDP-v4   | Ant-v4   |HalfCheetah-v4|Hopper-v4|Walker2d-v4|Average|
> | -------- | :--------: | :--------: | :--------: | :--------: | :--------: | :--------: |
> |SOPs (1k)| 117.1 |	158.2 |	174.9 	|89.1 	|180.9 	|144.0 |
> |Energy Consumptions ($nJ$)|9.01|12.18|13.46|6.86|13.93|11.09|
>
>
> **Comparison between ANNs running on NPUs and SNNs.** Table R5 concludes the estimated inference consumption by summarizing the results in Table R2, R3, R4.
>
> **Table R5:** Inference energy consumption for ANNs running on NPUs and SNNs, where the energy unit is nano-joule (nJ).
>
> | Method   | IDP-v4   | Ant-v4   |HalfCheetah-v4|Hopper-v4|Walker2d-v4|Average|
> | -------- | :--------: | :--------: | :--------: | :--------: | :--------: | :--------: |
> |ANN|272.33|295.60|283.41|274.26|283.41|281.78|
> |LIF|8.14|11.78|15.13|7.21|18.82|12.21|
> |PT-LIF|9.01|12.18|13.46|6.86|13.93|**11.09**|
>
> Table R5 demonstrates that SNN models consume less energy than the ANN models running on NPUs.
>
>
> Thank you again for your suggestion. We will include the energy consumption data for NPUs, based on the information provided in [2], in Table 2 of the manuscript. Please feel free to let us know if you have any remaining concerns.
>
>
> **References**
>
> [1] Ali, Asmer Hamid, Mozhgan Navardi, and Tinoosh Mohsenin. "Energy-Aware FPGA Implementation of Spiking Neural Network with LIF Neurons." arXiv preprint arXiv:2411.01628 (2024).
>
> [2] Millar, Josh, et al. "Benchmarking Ultra-Low-Power $\mu$NPUs." arXiv preprint arXiv:2503.22567 (2025).
>
> [3] Davies, Mike. "Taking neuromorphic computing to the next level with Loihi2." Intel Labs’ Loihi 2.1 (2021).
>
> [4] Merolla, Paul A., et al. "A million spiking-neuron integrated circuit with a scalable communication network and interface." Science 345.6197 (2014): 668-673.
>
> [5] Qiao, Ning, et al. "A reconfigurable on-line learning spiking neuromorphic processor comprising 256 neurons and 128K synapses." Frontiers in neuroscience 9 (2015): 141.
>
> [6] Hu, Yangfan, Huajin Tang, and Gang Pan. "Spiking deep residual networks." IEEE Transactions on Neural Networks and Learning Systems 34.8 (2021): 5200-5205.

---

### Official Review · Reviewer_JUY9 · 2025-06-30

**Clarity:** 3
**Significance:** 4
**Originality:** 4
**Rating:** 5
**Confidence:** 5

**Summary:**

This paper introduces a Proxy Target in the SNN-RL training framework to bridge the gap between the discrete nature of SNNs and the continuous control RL. While traditional RL algorithms rely on smooth target network updates, it conflicts with the non-differentiable dynamics of SNNs, leading to unstable training. The proposed method incorporates a differentiable proxy actor used only during training to stabilize learning while preserving the energy efficiency of SNNs during inference. Various experiments demonstrate significant performance gains, while the average performance across different environments even surpasses standard ANN baselines with simple LIF neurons.

**Questions:**

1. The authors claimed that “Simple neuron makes the best.”, Could the authors discuss more about why simple neurons perform better in this specific situation?

2. There are typos and informal writings in the manuscript. For example, in Figure 1, “inplicit” should be “implicit”; “Multi Iterations” should be “Multi-iteration” or “Multiple Iterations”.

**Ethical Concerns:**

["NO or VERY MINOR ethics concerns only"]

**Final Justification:**

The authors have successfully addressed all my concerns. It is remarkable that SNNs outperform ANNs in reinforcement learning tasks, which expands the application scope of SNNs beyond their well-known energy efficiency advantages. I will maintain my recommendation for acceptance.

**Limitations:**

See weaknesses.

**Paper Formatting Concerns:**

None.

**Quality:**

3

**Strengths And Weaknesses:**

Strengths
1. This paper addresses the critical challenge of training instability in SNN-based reinforcement learning by introducing a well-motivated proxy target framework that effectively mitigates the mismatch between discrete spiking dynamics and continuous control requirements.
2. The experimental evaluation is thorough, covering multiple environments and incorporating different neuron types (LIF, CLIF, ...). Results consistently show improved performance, reduced variance, and, notably the SNN-based RL outperforms ANN baselines. Moreover, the study also demonstrates the energy efficiency of SNNs by estimating inference energy consumption.
3. It is impressive to see SNNs outperform ANN in RL field.

Weaknesses
1. While the proxy actor effectively stabilizes reinforcement learning with an SNN actor, the additional training cost introduced by the implicit update mechanism is not thoroughly discussed. As illustrated in Figure 1, the implicit update appears to involve multiple backpropagation steps per iteration, in contrast to the single-step soft parameter update in traditional frameworks. This may lead to increased training time and computational overhead.

2. In line 212, the authors state that the proxy is a multilayer feedforward network. However, the effect of specific architectural choices (such as the number of layers and hidden units) is not clearly justified and studied. Additionally, the effect of the number of proxy update iterations (hyperparameter K) is not explored.

3. In the discussion on energy consumption, it would be beneficial to report the number of SOPs alongside the estimated energy values, as SOPs offer a more general measure of computational complexity. Since energy estimates can vary significantly depending on the hardware, including a comparison of SOPs would enhance the generalizability of the energy efficiency claims.

---

> ### Author Rebuttal · Authors · 2025-07-31
>
> We sincerely appreciate your recognition of our work. We are delighted that you noted that "It is impressive to see SNNs outperform ANN in RL field". We will strive to address your relevant questions.
>
>
> >While the proxy actor effectively stabilizes reinforcement learning with an SNN actor, the additional training cost introduced by the implicit update mechanism is not thoroughly discussed. As illustrated in Figure 1, the implicit update appears to involve multiple backpropagation steps per iteration, in contrast to the single-step soft parameter update in traditional frameworks. This may lead to increased training time and computational overhead.
>
>
> Thank you for this valuable question. To reduce the training cost associated with multiple backpropagation steps during the proxy update, we can adopt a one-step proxy update setting for LIF neuron, with the proxy update iteration $K=1$ and proxy learning rate $lr_{proxy}=3\cdot10^{-3}$ (where the default setting is $K=3$ and $lr_{proxy}=3\cdot10^{-3}$). This lightweight configuration significantly reduces computation without major performance loss (only a 0.4% performance degradation).
>
> **Table R1:** Maximum average returns and average performance gain (APG) against ANN with different settings of the proposed PT-LIF.
> | Method                         | IDP-v4          | Ant-v4          | HalfCheetah-v4  | Hopper-v4      | Walker2d-v4    | APG     |
> | ------ | :--------: | :--------: | :--------: | :--------: | :--------: | :------: |
> | ANN                            | $7503 \pm 3713$ | $4770 \pm 1014$ | $10857 \pm 475$ | $3410 \pm 164$ | $4340 \pm 383$ | $0.00$% |
> | PT-LIF (default)               | $9348 \pm 1$    | $5383 \pm 250$  | $10103 \pm 607$ | $3385 \pm 157$ | $4314 \pm 423$ | $5.84$% |
> | PT-LIF (one-step proxy update) | $9348 \pm 0$    | $5032 \pm 535$  | $9440 \pm 553$  | $3504 \pm 159$ | $4661 \pm 263$ | $5.44$% |
>
>
> We further measured the training time required for 100,000 steps on the Ant-v4 environment using an RTX 3090 GPU and an Intel Xeon Platinum 8362 CPU. We then computed the relative training time compared to the baseline vanilla LIF model.
>
> **Table R2:** The relative training time (compared with vanilla LIF) and the average performance gain (against ANN) across different SNN models.
> ||Vanilla LIF|MDC-SAN|ILC-SAN|PT-LIF (one-step proxy update)\*|PT-LIF (default)|
> | -------- | :--------: | :--------: | :--------: | :--------: | :--------: |
> |Relative Training Time|1.00|1.41|1.79|**1.35**|2.24|
> |Average Performance Gain|-10.54%|0.37%|4.64%|5.44%|**5.84%**|
>
>
> The trade-off between training cost and performance is a long-standing issue. All SOTA SNN algorithms require more training time than the vanilla LIF model. However, our PT-LIF achieves the highest performance gain with reasonable training overhead. Notably, with a lightweight one-step proxy update configuration, our PT-LIF requires even less training time than other SOTA models (e.g. MDC-SAN, ILC-SAN), while still surpassing them in terms of performance.
>
>
> In addition, our model remains inference efficient. In contrast, other SNN variants (e.g. MDC-SAN, ILC-SAC) introduce architectural or neuronal complexities that increase inference energy consumption.
>
>
> >In line 212, the authors state that the proxy is a multilayer feedforward network. However, the effect of specific architectural choices (such as the number of layers and hidden units) is not clearly justified and studied. Additionally, the effect of the number of proxy update iterations (hyperparameter K) is not explored.
>
> Thank you for this valuable suggestion. We conduct further experiments on the effects of proxy update iterations (K), the depth and width of proxy architectures.
>
> **Table R3:** Max average returns and average performance gain (APG) against the default settings.
> ||Ant-v4|HalfCheetah-v4|Hopper-v4|Walker2d-v4|APG|
> | -------- | :--------: | :--------: | :--------: | :--------: | --------: |
> |Without Proxy|$4294 \pm 1170$|$9404 \pm 625$|$3520 \pm 94$|$1862 \pm 1450$|$-20.01$%|
> |K=2|$5532 \pm 2714$|$8461 \pm 4213$|$3419 \pm 1677$|$4229 \pm 2083$|$-3.62$%
> |K=3|$5383 \pm 250$|$10103 \pm 607$|$3385 \pm 157$|$4314 \pm 423$|$0.00$%|
> |K=5|$5587 \pm 2745$|$9750 \pm 4782$|$2374 \pm 1648$|$4131 \pm 2033$|$-8.46$%|
> |depth=1|$5062 \pm 960$|$9272 \pm 625$|$3453 \pm 141$|$4170 \pm 340$|$-3.89$%
> |depth=2|$5383 \pm 250$|$10103 \pm 607$|$3385 \pm 157$|$4314 \pm 423$|$0.00$%|
> |depth=3|$5176 \pm 2559$|$9728 \pm 4795$|$3435 \pm 1687$|$4067 \pm 2018$|$-2.96$%
> |width=(256,256)|$4491 \pm 2424$|$8464 \pm 4177$|$3475 \pm 1708$|$4242 \pm 2118$|$-7.95$%
> |width=(512,512)|$5383 \pm 250$|$10103 \pm 607$|$3385 \pm 157$|$4314 \pm 423$|$0.00$%|
> |width=(800,600)|$5459 \pm 2675$|$9616 \pm 4734$|$3495 \pm 1715$|$2841 \pm 1715$|$-7.26$%|
>
> These results suggest that the proposed proxy network is robust to moderate variations in hyperparameters, as all these settings could outperform the baseline without the proxy. The default configuration (K=3, depth=2, width=(512,512)) achieves strong and stable performance across different tasks. Meanwhile, setting these parameters too low or too high tends to degrade performance.
>
> >In the discussion on energy consumption, it would be beneficial to report the number of SOPs alongside the estimated energy values, as SOPs offer a more general measure of computational complexity. Since energy estimates can vary significantly depending on the hardware, including a comparison of SOPs would enhance the generalizability of the energy efficiency claims.
>
> Thank you for the valuable suggestion. We show the number of SOPs in Table R4, the proposed method also has low SOPs on average.
>
> **Table R4:** The average number of SOPs per inference in different environments across $5$ random seeds. Where the unit is $1,000$ SOPs.
> | Method   | IDP-v4   | Ant-v4   |HalfCheetah-v4|Hopper-v4|Walker2d-v4|Average|
> | -------- |:--------: | :--------: | :--------: | :--------: | :--------: | :--------: |
> |LIF   |105.8 |	152.9 |	196.6 	|93.6 	|244.4 |	158.6|
> |PT-LIF| 117.1 |	158.2 |	174.9 	|89.1 	|180.9 	|144.0 |
>
> We will add the number of SOPs alongside the estimated energy values in the energy consumption part of the main text to offer a more general measure of computational complexity.
>
>
> >The authors claimed that “Simple neuron makes the best.”, Could the authors discuss more about why simple neurons perform better in this specific situation?
>
> Thank you for this insightful question. Once the SNN surpasses the ANN in performance, the training bottleneck shifts from the capacity of the SNN actor to the reinforcement learning algorithm itself. Therefore, incorporating more complex spiking neurons may introduce unnecessary training complexity, potentially degrading performance.
>
>
> >There are typos and informal writings in the manuscript. For example, in Figure 1, “inplicit” should be “implicit”; “Multi Iterations” should be “Multi-iteration” or “Multiple Iterations”.
>
> Thanks for your detailed review of the manuscript. We will correct these errors and thoroughly revise the manuscript.

---

> > ### Comment · Reviewer_JUY9 · 2025-08-06
> >
> > I appreciate the thorough reply and additional experiments. The authors have successfully addressed all my concerns. It is remarkable that SNNs outperform ANNs in reinforcement learning tasks, which expands the application scope of SNNs beyond their well-known energy efficiency advantages. I will maintain my recommendation for acceptance.

---

> ### Author Response · Authors · 2025-08-06
>
> We sincerely thank you for recognizing our work. We are encouraged by your acknowledgment of the superior performance compared to ANNs. Please feel free to let us know if you have any further questions.

---

### Official Review · Reviewer_i6Dh · 2025-07-03

**Clarity:** 3
**Significance:** 3
**Originality:** 3
**Rating:** 4
**Confidence:** 2

**Summary:**

This paper studies the challenge of applying continuous-control reinforcement learning algorithms to Spiking Neural Networks (SNNs), pointing out that standard RL mechanisms—especially the soft update of target networks—are incompatible with the discrete, non-differentiable nature of SNN spikes. The authors propose a proxy target framework in which a continuous, differentiable proxy network is used during training to smooth updates and bridge the gap between SNNs and RL algorithms originally designed for ANNs. Experiments demonstrate that SNNs trained with this framework achieve improved stability and up to 32% higher average performance in standard MuJoCo environments, even surpassing traditional ANNs in some settings. The framework preserves the energy-efficiency benefits of SNNs at inference.

**Questions:**

n/a

**Ethical Concerns:**

["NO or VERY MINOR ethics concerns only"]

**Final Justification:**

Thanks for the rebuttal. Combining the comments with other reviewers, I will maintain my original score.

**Limitations:**

Yes

**Quality:**

3

**Strengths And Weaknesses:**

Strengths:
- Identification of a Previously Unaddressed Limitation: The paper provides a clear diagnosis of a previously underappreciated source of instability in SNN-based RL for continuous control—the incompatibility of non-differentiable spiking output with soft target updates. This is demonstrated empirically (see Figure 2b), which shows abrupt, discrete changes in SNN target outputs that undermine stable learning.

- Methodological Contribution: The proposed "proxy target" mechanism is simple yet effective. By using a continuous, differentiable actor only for target updates during training, it permits smooth transitions needed for stable RL optimization without sacrificing SNN deployment benefits. The update mechanism is derived clearly, and implementation details are provided.

- Comprehensive Experimental Validation: The authors provide thorough empirical evidence across five MuJoCo environments, using multiple spiking neuron types (LIF, CLIF, DN). Learning curves (Figure 4) illustrate consistent improvement in convergence speed and final return, with shaded regions for statistical reliability. Table 1 compares max average returns to multiple strong SNN baselines and ANN-based RL, showing the proposed PT-LIF outperforms all baselines in average performance gain (5.84%).

Weaknesses:
- Limited Theoretical Analysis: While the mechanism is well-motivated and empirically validated, the manuscript lacks rigorous theoretical support. The smoothing effect of the proxy is justified mainly by intuition and plotting, without formal stability/convergence claims (see Section 4, pages referencing Figures 2c-d). This limits broader generalizability to other RL algorithms or network types.

- Ablation Shortcomings: The experimental section would benefit from ablations isolating the effects of proxy update frequency, the batch size (N) used for proxy training, or the depth/width of proxy architectures to clarify the mechanism’s robustness and optimal configurations. While Figure 2c-d and Figure 4 highlight core effects, missing such ablations leaves open questions about sensitivity and practical guidance.

- Missing Key Baselines: Although several state-of-the-art SNN RL methods are compared in Table 1, additional direct baselines such as SNNs with architectural/topological trims but without proxy, or variants using different activation surrogates, would further clarify the gain is from the proxy and not other confounds.

- Proxy-Online SNN Gap Quantification: While Figure 2d visually indicates the proxy network closely tracks the SNN actor, a quantitative, environment-wide measure of this approximation gap is not provided. Including such a metric (e.g., proxy-SNN output distance during learning) across seeds/environments would clarify how tightly the proxy must follow the SNN to ensure learning stability.

---

> ### Author Rebuttal · Authors · 2025-07-31
>
> We would like to thank you for your constructive questions and suggestions! We will conduct a detailed clarification and discussion for your concerns.
>
> >Limited Theoretical Analysis: While the mechanism is well-motivated and empirically validated, the manuscript lacks rigorous theoretical support. The smoothing effect of the proxy is justified mainly by intuition and plotting, without formal stability/convergence claims (see Section 4, pages referencing Figures 2c-d). This limits broader generalizability to other RL algorithms or network types.
>
> Thank you for the valuable suggestion. Here we provide a theoretical analiysis demonstrating that the proxy network enables smoother policy updates than the SNN target.
>
> **Theorem 1:** For proxy network $\pi_{\phi'}^{Proxy}$ updated by minimizing the loss $L_{proxy}$ in Eq.11, during each update, as the proxy learning rate $lr_{proxy}$ approaches to zero, $\left\|\pi_{\phi_{new}'}^{Proxy}(s)-\pi_{\phi_{old}'}^{Proxy}(s)\right\|$ also approches to zero. Where $\phi_{old}'$ and $\phi_{new}'$ are the parameters of the proxy network before and after the update, respectively.
>
> **Proof 1:** By standard gradient descent, we have:
> $$\lim_{lr_{proxy}\to0}\left\|(\phi_{new}'-\phi_{old}')\right\|=\lim_{lr_{proxy} \to 0} \left \|lr_{proxy} \cdot \nabla_{\phi_{old}'} L_{proxy}\right\|=0.$$
> Under the assumption that $\pi_{\phi'}^{Proxy}(s)$ is continuously differentiable with respect to $ϕ'$, we apply a first-order Taylor expansion:
> $$\lim_{lr_{proxy}\to0}\left\|\pi_{\phi_{new}'}^{Proxy}(s)-\pi_{\phi_{old}'}^{Proxy}(s)\right\|= \lim_{lr_{proxy}\to0}\left\|(\phi_{new}'-\phi_{old}')\nabla_{\phi'}\pi_{\phi_{old}'}^{Proxy}(s)\right\|=0.$$
>
> This result shows that the proxy network supports sufficiently small policy updates. In contrast, due to the discrete nature of SNN target, it cannot produce sufficiently small output changes. Thus, the smoothing effect of proxy is superior, which enables smooth and stable optimization.
>
> We will include the above theoretical analysis in the main text after Section 4.2.
>
> >Ablation Shortcomings: The experimental section would benefit from ablations isolating the effects of proxy update frequency, the batch size (N) used for proxy training, or the depth/width of proxy architectures to clarify the mechanism’s robustness and optimal configurations. While Figure 2c-d and Figure 4 highlight core effects, missing such ablations leaves open questions about sensitivity and practical guidance.
>
> Thank you for this valuable question. We conducts ablations isolating the effects of proxy update iterations (K), the batch size (N) used for proxy training, the depth and width of proxy architectures.
>
> **Table R1:** Maximum average returns and average performance gain (APG) compared to the default settings.
> ||Ant-v4|HalfCheetah-v4|Hopper-v4|Walker2d-v4|APG|
> | -------- | :--------: | :--------: | :--------: | :--------: | --------: |
> |Without Proxy|$4294 \pm 1170$|$9404 \pm 625$|$3520 \pm 94$|$1862 \pm 1450$|$-20.01$%|
> |K=2|$5532 \pm 2714$|$8461 \pm 4213$|$3419 \pm 1677$|$4229 \pm 2083$|$-3.62$%
> |K=3|$5383 \pm 250$|$10103 \pm 607$|$3385 \pm 157$|$4314 \pm 423$|$0.00$%|
> |K=5|$5587 \pm 2745$|$9750 \pm 4782$|$2374 \pm 1648$|$4131 \pm 2033$|$-8.46$%
> |N=128|$4967 \pm 1086$|$9688 \pm 786$|$3517 \pm 106$|$3521 \pm 1479$|$-6.59$%
> |N=256|$5383 \pm 250$|$10103 \pm 607$|$3385 \pm 157$|$4314 \pm 423$|$0.00$%|
> |N=512|$5374 \pm 2641$|$10064 \pm 4931$|$2380 \pm 1655$|$2940 \pm 1775$|$-15.53$%
> |depth=1|$5062 \pm 960$|$9272 \pm 625$|$3453 \pm 141$|$4170 \pm 340$|$-3.89$%
> |depth=2|$5383 \pm 250$|$10103 \pm 607$|$3385 \pm 157$|$4314 \pm 423$|$0.00$%|
> |depth=3|$5176 \pm 2559$|$9728 \pm 4795$|$3435 \pm 1687$|$4067 \pm 2018$|$-2.96$%
> |width=(256,256)|$4491 \pm 2424$|$8464 \pm 4177$|$3475 \pm 1708$|$4242 \pm 2118$|$-7.95$%
> |width=(512,512)|$5383 \pm 250$|$10103 \pm 607$|$3385 \pm 157$|$4314 \pm 423$|$0.00$%|
> |width=(800,600)|$5459 \pm 2675$|$9616 \pm 4734$|$3495 \pm 1715$|$2841 \pm 1715$|$-7.26$%|
>
> These results suggest that the proposed proxy network is robust to moderate variations in hyperparameters and consistently outperforms the baseline without the proxy. The default configuration (K=3, N=256, depth=2, width=(512,512)) achieves strong and stable performance across different tasks. Meanwhile, setting these parameters too low or too high tends to degrade performance.
>
>
> >Missing Key Baselines: Although several state-of-the-art SNN RL methods are compared in Table 1, additional direct baselines such as SNNs with architectural/topological trims but without proxy, or variants using different activation surrogates, would further clarify the gain is from the proxy and not other confounds.
>
> Thanks for the suggestion. We added several new baselines. For SNNs with different architects, we conducted experiments using vanilla LIF neuron, pop-SAN[1], MDC-SAN[2]. For SNNs with topological modifications, we conducted experiments of ILC-SAN[3]. For SNN with different surrogates, we  added experiments on LIF neuron with tanh surrogate function without proxy. In addition, we also added a new baseline for converting the pre-trained ANN to an SNN (using the method proposed in [4] with 100 time steps).
>
> **Table R2:** Max average returns and average performance gain (APG) against ANN.
>
> |Method|IDP-v4|Ant-v4|HalfCheetah-v4|Hopper-v4|Walker2d-v4|APG|
> | -------- | :--------: | :--------: | :--------: | :--------: | :--------: |--------: |
> |ANN|$7503 \pm 3713$|$4770 \pm 1014$|$10857 \pm 475$|$3410 \pm 164$|$4340 \pm 383$|$0.00$%|
> |ANN-SNN|$3859\pm4440$|$3550\pm963$|$8703\pm658$|$3098\pm281$|$4235\pm354$|$-21.11$%|
> |Vanilla LIF with rectangular surrogate|$9347 \pm 1$|$4294 \pm 1170$|$9404 \pm 625$|$3520 \pm 94$|$1862 \pm 1450$|$-10.54$%|
> |Vanilla LIF with tanh surrogate|$9348 \pm 1$|$4486 \pm 2380$|$9019 \pm 4436$|$3361 \pm 1654$|$4147 \pm 2036$|$-0.83$%|
> |pop-SAN|$9351 \pm 1$|$4590 \pm 1006$|$9594 \pm 689$|$2772 \pm 1263$|$3307 \pm 1514$|$-6.66$%|
> |MDC-SAN|$9350 \pm 1$|$4800 \pm 994$|$9147 \pm 231$|$3446 \pm 131$|$3964 \pm 1353$|$0.37$%|
> |ILC-SAN|$9352 \pm 1$|$5584 \pm 272$|$9222 \pm 615$|$3403 \pm 148$|$4200 \pm 717$|$4.64$%|
> |PT-CLIF|$9351 \pm 1$|$5014 \pm 1074$|$9663 \pm 426$|$3526 \pm 112$|$4564 \pm 555$|$5.46$%|
> |PT-DN|$9350 \pm 1$|$5400 \pm 277$|$9347 \pm 666$|$3507 \pm 144$|$4277 \pm 650$|$5.06$%|
> |PT-LIF|$9348 \pm 1$|$5383 \pm 250$|$10103 \pm 607$|$3385 \pm 157$|$4314 \pm 423$|$5.84$%|
>
> The proposed algorithms (i.e. PT-CLIF, PT-DN, and PT-LIF) still outperform other baselines.
>
> We will include the new results in Table 1 of the main text to further clarify the gains from the proxy.
>
>
> >Proxy-Online SNN Gap Quantification: While Figure 2d visually indicates the proxy network closely tracks the SNN actor, a quantitative, environment-wide measure of this approximation gap is not provided. Including such a metric (e.g., proxy-SNN output distance during learning) across seeds/environments would clarify how tightly the proxy must follow the SNN to ensure learning stability.
>
> Thank you for the suggestion. We evaluated the mean squared output error between the target network and the online SNN during training (with the same inputs), and averaged them over 4 different environments (i.e., Ant-v4, HalfCheetah-v4, Hopper-v4, and Walker2d-v4) over 5 random seeds.  The average approximation gap is relatively smaller with the proxy network in all training steps, demonstrating that the proxy can more closely follow the SNN to ensure learning stability.
>
> **Table R3:** The average approximation gap between the target network and the SNN actor in different training steps.
> |Method|$0.2$M|$0.4$M|$0.6$M|$0.8$M|$1.0$M|
> | -------- |:--------: | :--------: | :--------: | :--------: | :--------: |
> |LIF|0.082|0.067|0.062|0.067|0.060|
> |PT-LIF|0.079|0.067|0.053|0.051|0.052|
>
>
> **References**
>
>
> [1] Guangzhi Tang, Neelesh Kumar, Raymond Yoo, and Konstantinos Michmizos. Deep reinforcement learning with population-coded spiking neural network for continuous control. In Conference on Robot Learning, pages 2016–2029. PMLR, 2021.
>
> [2] Duzhen Zhang, Tielin Zhang, Shuncheng Jia, and Bo Xu. Multi-sacle dynamic coding improved spiking actor network for reinforcement learning. In Proceedings of the AAAI conference on artificial intelligence, volume 36, pages 59–67, 2022.
>
> [3] Ding Chen, Peixi Peng, Tiejun Huang, and Yonghong Tian. Fully spiking actor network with intralayer connections for reinforcement learning. IEEE Transactions on Neural Networks and Learning Systems, 36(2):2881–2893, 2024a.
>
> [4] Tong Bu, Maohua Li, and Zhaofei Yu. Inference-Scale Complexity in ANN-SNN Conversion for High-Performance and Low-Power Applications. In Proceedings of the Computer Vision and Pattern Recognition Conference, pages 24387-24397, 2025.

---

### Official Review · Reviewer_PQzp · 2025-07-05

**Clarity:** 1
**Significance:** 2
**Originality:** 2
**Rating:** 2
**Confidence:** 4

**Summary:**

This paper solves a critical mismatch between SNNs and continuous control RL: SNNs' discrete spikes clash with RL's need for smooth optimization, causing training instability. This paper introduces the Proxy Target Framework to resolve this. During training, a continuous network (the proxy) replaces the SNN target network, providing the smooth updates necessary to stabilize learning. Crucially, the efficient SNN is still used during inference, preserving energy benefits. This approach significantly boosts stability and performance.

**Questions:**

see weakness (W1-W4)

**Ethical Concerns:**

["NO or VERY MINOR ethics concerns only"]

**Limitations:**

Increased Training Overhead & Limited Task Generality

**Quality:**

2

**Strengths And Weaknesses:**

**Strengths:**
1. This paper pinpoints the core conflict between discrete SNN dynamics and the continuous soft-update mechanisms in RL.
2. By using the proxy only during training, it stabilizes the learning process of SNNs.


**Weaknesses:**
W1. The auxiliary proxy network increases computational cost and complexity during the training phase, even if inference remains efficient. This overhead should be analyzed.

W2. The solution is tailored for Off-policy RL algorithms using target networks (e.g., DDPG, TD3). Its applicability to On-policy methods (e.g., PPO) is unclear.

W3. Achieving performance superior to ANNs might require high spike firing rates or many timesteps, potentially diminishing the claimed energy efficiency benefits (sparsity and latency).

W4. The mechanism for updating the proxy network to match the SNN likely introduces significant interaction costs between the two models. This dual update process could become a training bottleneck and requires further discussion on optimization.

---

> ### Author Rebuttal · Authors · 2025-07-31
>
> We are encouraged that you recognize the effectiveness of the proposed proxy target in terms of training stability and performance. We will do our best to address your concerns in detail.
>
> >W1. The auxiliary proxy network increases computational cost and complexity during the training phase, even if inference remains efficient. This overhead should be analyzed.
>
>
> Thank you for this suggestion. To quantify the computational cost during training, we measured the training time required for 100,000 steps on the Ant-v4 environment using an RTX 3090 GPU and an Intel Xeon Platinum 8362 CPU. We then computed the relative training time compared to the baseline vanilla LIF model.
>
> **Table R1:** The relative training time (compared with vanilla LIF) and the average performance gain (against ANN) across different SNN models.
> ||Vanilla LIF|MDC-SAN|ILC-SAN|PT-LIF (one-step proxy update)\*|PT-LIF (default)|
> | -------- | :--------: | :--------: | :--------: | :--------: | :--------: |
> |Relative Training Time|1.00|1.41|1.79|**1.35**|2.24|
> |Average Performance Gain|-10.54%|0.37%|4.64%|5.44%|**5.84%**|
>
>
> The trade-off between training cost and performance is a long-standing issue. All SOTA SNN algorithms require more training time than the vanilla LIF model. However, our PT-LIF achieves the highest performance gain with reasonable training overhead. Notably, with a lightweight one-step proxy update configuration, our PT-LIF requires even less training time than other SOTA models (e.g. MDC-SAN, ILC-SAN), while still surpassing them in terms of performance.
>
>
> In addition, our model remains inference efficient. In contrast, other SNN variants (e.g. MDC-SAN, ILC-SAC) introduce architectural or neuronal complexities that increase inference energy consumption.
>
>
> \*: To reduce training costs, we can adopt a one-step proxy update setting for LIF neuron, with the proxy update iteration $K=1$ and proxy learning rate $lr_{proxy}=3\cdot10^{-3}$ (where the default setting is $K=3$ and $lr_{proxy}=3\cdot10^{-3}$). This lightweight configuration significantly reduces computation without major performance loss. Please refer to Table R1 in our response to Reviewer JUY9 for detailed performance under this setting across all environments.
>
> >W2. The solution is tailored for Off-policy RL algorithms using target networks (e.g., DDPG, TD3). Its applicability to On-policy methods (e.g., PPO) is unclear.
>
> Thank you for this insightful question. This work focuses on resolving the conflict between smooth target network updates and the non-differentiable dynamics of spiking neural networks (SNNs), a challenge that arises specifically in off-policy reinforcement learning (RL) algorithms such as DDPG and TD3, which employ target networks. Since on-policy algorithms like PPO do not use target networks, the proposed solution does not directly apply to them and falls outside the current scope of our study. We will revise the introduction to explicitly state that the proposed approach is tailored for off-policy algorithms.
>
> We would also like to highlight that off-policy algorithms remain a central focus in RL research due to their better sample efficiency, as they can reuse past experience from the replay buffer. Hence, we believe that addressing the challenges of applying SNNs in these algorithms is of great practical value.
>
> Moreover, adapting on-policy methods to SNNs also has its own challenges. For example, methods like PPO rely on smooth policy updates, while the discrete nature of SNNs can lead to abrupt output shifts in the policy network that may destabilize training. We are interested in exploring potential solutions in our future work.
>
>
> > W3. Achieving performance superior to ANNs might require high spike firing rates or many timesteps, potentially diminishing the claimed energy efficiency benefits (sparsity and latency).
>
>
> Thank you for this insightful comment. We would like to clarify that achieving performance superior to ANNs does not necessarily require higher spike firing rates or longer timesteps in our method. To support this point, we report the average firing rates in Table R2. The firing rate of our model remains within a typical range and is even lower than that of the baseline SNN in most cases.
>
> **Table R2:** Comparison of the average firing rates of spiking neurons in different environments over $5$ random seeds.
>
> | Method   | IDP-v4   | Ant-v4   |HalfCheetah-v4|Hopper-v4|Walker2d-v4|Average|
> | -------- | :--------: | :--------: | :--------: | :--------: | :--------: | :----:|
> |LIF|0.22|0.31|0.40|0.25|0.47|0.33|
> |PT-LIF|0.24|0.33|0.37|0.25|0.40|**0.32**|
>
> Furthermore, we would like to clarify that the number of timesteps is consistently set to 5 across all experiments, including those for the baseline SNN and other existing algorithms.
>
> These results indicate that our method maintains moderate spike activity and low latency while outperforming ANN baselines. Furthermore, as shown in Table 2 of the main manuscript, our model achieves lower energy consumption than both the baseline SNN and ANN models, preserving energy efficiency.
>
> To avoid any potential misunderstanding, we will revise the manuscript to explicitly emphasize the timestep setting and include the firing rate statistics in the energy analysis section of the manuscript (Section 5).
>
>
> >W4. The mechanism for updating the proxy network to match the SNN likely introduces significant interaction costs between the two models. This dual update process could become a training bottleneck and requires further discussion on optimization.
>
> Thank you for this insightful question. To address the concern clearly, we would like to respond from three perspectives (energy efficiency, optimization performance, and clarification of the dual update process).
>
> * Energy Efficiency:
> We would like to emphasize that the proposed framework requires less inference energy compared to the ANN or SNN baseline (as shown in Table 2 of the main manuscript), and thus remains inference efficient. Moreover, the proposed method introduces reasonable additional computational overhead during training. As discussed in our response to Question W1, the overall training cost remains comparable to existing methods.
> * Optimization Performance:
> We would like to clarify that the proxy network is not a training bottleneck in terms of performance. On the contrary, it is introduced to alleviate a key bottleneck observed in traditional SNN-RL frameworks. Specifically, we found that directly using an SNN as the target network can lead to unstable updates due to its discrete and abrupt output dynamics, which hinders optimization. The proxy network helps to smooth the learning targets and improves the stability and performance of the learning process. For theoretical justification, please refer to our response to Reviewer i6Dh, Question 1. We also showed empirical evidence supporting the effectiveness of the proxy network in the experiments section of the main manuscript.
> * Clarification of the Dual Update Process:
> The presence of a dual update process in our framework is consistent with standard actor-critic RL algorithms such as DDPG and TD3. These algorithms contain actors, critics, and corresponding target networks (or proxy targets). In our method, the only distinction lies in how we update the target actor (proxy). Instead of performing a soft update for the SNN target actor, we introduce an implicit update from the proxy target to the SNN actor. Importantly, the actor network remains unaffected during this process, as its parameters are frozen. This update scheme is fully compatible with the conventional actor-critic paradigm and has been empirically shown to improve training stability and overall performance.
>
> We will revise the manuscript to include a more detailed explanation of this mechanism. Please feel free to let us know if there are any remaining concerns.

---

> > ### Comment · Reviewer_PQzp · 2025-08-06
> >
> > We appreciate the authors' detailed explanation. However, the response still does not fully address our concerns. Specifically, we find the reported training cost of 100,000 steps to be disproportionately high for SNN models. Unlike conventional ANNs, longer training does not necessarily guarantee better convergence or performance in SNNs, especially given the sensitivity of spike-based learning to hyperparameters and temporal dynamics. Moreover, it remains unclear how the authors ensure convergence within 100,000 steps—no learning curve or convergence analysis is provided. Regarding hardware feasibility, we question whether a single RTX 3090 can realistically support such training, given that the memory and computational load in SNNs heavily depends on the time window, which appears non-trivial in this work. Finally, while the proposed scheme demonstrates potential improvements in theory, it is not clear whether these benefits translate into practical gains under real deployment constraints. We suggest that the authors provide more rigorous empirical evidence, including convergence curves, resource utilization reports, and comparisons under realistic training budgets.

---

> ### Author Response · Authors · 2025-08-06
>
> We sincerely thank you for your thoughtful comments. There may be some potential misunderstandings, and we are grateful for the opportunity to clarify.
>
> 1. Training Steps Clarification
> The mention of “100,000 training steps” in our previous response was solely for reporting the relative training time cost in the second row of Table R1 (due to the limited time of the rebuttal period, it is only tested for 100,000 training steps). All performance evaluations (i.e., maximum average return, average performance gain) throughout the rebuttals (including Table R1) are based on 1,000,000 RL training steps, consistent with Figure 4 in the original manuscript (except for the InvertedDoublePendulum-v4 task, where 50,000 training steps were used due to early convergence).
> 2. Clarification on the RL Training Steps and the SNN Simulation Time Steps
> We would like to emphasize that the 1,000,000 RL steps refer to the number of training iterations in the RL training process, not to the internal SNN simulation time steps. In all experiments, we use a fixed number of 5 SNN simulation time steps per inference. For each RL environment step, the SNN actor performs a single inference pass consisting of 5 simulation time steps, after which all internal neuron states (e.g., membrane potentials) are reset. Therefore, the SNN does not accumulate temporal dynamics beyond a window of 5 time steps. To avoid further ambiguity, we will explicitly use "training steps" for RL training iterations and "time steps" for the SNN simulation steps in the revised manuscript.
>
>
>
> We apologize for any potential misunderstandings or confusion. We will do our best to address your concerns in detail.
>
>
> > Specifically, we find the reported training cost of 100,000 steps to be disproportionately high for SNN models.
>
> We understand your concern regarding the training cost, and we would like to clarify that the training overhead is acceptable. As shown in Table R1, the proposed PT-LIF model (with one-step proxy update) incurs a training time cost of only 1.35× relative to the vanilla LIF baseline. This increase falls within a reasonable range and not significantly higher than typical overhead for SNN training. Moreover, PT-LIF is more efficient than several state-of-the-art SNN-based algorithms, such as MDC-SAN[3] (1.41×) and ILC-SAN[4] (1.79×), while also achieving superior performance than MDC-SAN and ILC-SAN.
>
> We acknowledge that training SNNs on GPUs often incurs higher computational overhead than ANNs, which remains an open research challenge in the SNN community. However, our model uses only 5 simulation time steps per inference, which helps control the training cost. In addition, it is important to highlight that SNNs offer significant energy savings during inference, especially when deployed on neuromorphic hardware. Thus, while training may be more expensive, SNNs remain highly energy-efficient during deployment.
>
> >Unlike conventional ANNs, longer training does not necessarily guarantee better convergence or performance in SNNs, especially given the sensitivity of spike-based learning to hyperparameters and temporal dynamics.
>
> In terms of convergence and performance, we would like to clarify that the proposed method is superior. As shown in Figure 4 and Table 1 of the manuscript, our method consistently outperforms existing SNN-based RL algorithms and even surpasses conventional ANN-based approaches. This demonstrates the effectiveness of our design despite the inherent challenges of spike-based learning.
>
> Regarding hyperparameter sensitivity, we emphasize that no special tuning was performed for our SNN models. The architectural settings (e.g., network depth and width) and learning rate were directly adopted from standard ANN-based RL algorithm [1]. Furthermore, the neuron model and the surrogate gradient function were kept consistent with prior SNN works [2–4]. In addition, Table R1 in the response to reviewer i6Dh suggests that the proposed proxy network is robust to moderate variations in hyperparameters, as it consistently outperforms the baseline without the proxy. Therefore, the strong performance of our method is not the result of extensive hyperparameter optimization, but rather stems from the proposed learning framework.
>
> We hope this addresses the concern and clarifies that the convergence and performance of our method are achieved without excessive tuning.

---

> ### Author Response · Authors · 2025-08-06
>
> >Moreover, it remains unclear how the authors ensure convergence within 100,000 steps—no learning curve or convergence analysis is provided.
>
> We would like to clarify that we do **not** claim convergence within 100,000 training steps (neither SNNs nor ANNs). In fact, for most continuous control tasks (such as Ant-v4, HalfCheetah-v4, Hopper-v4, and Walker2d-v4), all statistical performance results reported are based on **1,000,000 training steps**, which is standard practice in RL benchmarks [1]. This includes both ANN and SNN variants. For detailed learning curves, please refer to Figure 4 and Figure 7 in the manuscript.
>
> The mention of “100,000 steps” in our earlier response was only to report the relative training time cost, as shown in Table R1. It does not refer to the total number of training iterations used for performance evaluation or convergence analysis. We apologize for any confusion this may have caused and will make this point clearer in the revised version of the manuscript.
>
> >Regarding hardware feasibility, we question whether a single RTX 3090 can realistically support such training, given that the memory and computational load in SNNs heavily depends on the time window, which appears non-trivial in this work.
>
> We appreciate your concern regarding hardware feasibility. We would like to clarify that the spiking actor network used in our work is configured with only **5 simulation time steps** and consists of two fully connected hidden layers with 256 neurons per layer. As a result, the computational and memory demands of our SNN are relatively modest—substantially lower than those of widely used models such as AlexNet, VGG16, or ResNet18.
>
> Empirically, training our proposed PT-LIF on a single RTX 3090 GPU results in approximately **21–24% GPU utilization** and consumes approximately **400 MiB of GPU memory**. In practice, we are able to train 10 agents simultaneously (across different environments and random seeds) on a single RTX 3090 without exceeding its computational capacity. Therefore, we confirm that a single RTX 3090 is sufficient to support the reported training setup.
>
> >Finally, while the proposed scheme demonstrates potential improvements in theory, it is not clear whether these benefits translate into practical gains under real deployment constraints.
>
> Thank you for this insightful question. To address deployment concerns, we provide inference energy consumption results in Table 2 of the manuscript. These results demonstrate that our proposed method is more energy-efficient than both baseline SNN and ANN models during inference, highlighting its potential advantages under deployment constraints.
>
> While real-world deployment on neuromorphic hardware would offer the most direct validation, current limitations in accessible neuromorphic platforms prevent us from implementing and benchmarking our model in such environments. We have acknowledged this limitation explicitly in the Limitations section of the manuscript. Nevertheless, the efficiency gains observed in our simulation-based evaluation suggest strong potential for practical deployment once suitable hardware becomes available.

---

> ### Author Response · Authors · 2025-08-06
>
> >We suggest that the authors provide more rigorous empirical evidence, including convergence curves, resource utilization reports, and comparisons under realistic training budgets.
>
> Thank you for your valuable suggestion.
> In terms of training curves, Figure 4 and Figure 7 in the manuscript shows the learning curves of the proposed PT-LIF model (with default settings) and other baselines (including ANNs and SNNs). As we cannot post PDF files, we further provide the following tables to show the average return for the  PT-LIF model (with one-step proxy update) compared with vanilla LIF. The  PT-LIF model (with one-step proxy update) achieves higher overall performance compared with the vanilla LIF baseline.
>
> Table C1: Average return during different training steps in the Ant-v4 environment.
> |Method|0M|0.2M|0.4M|0.6M|0.8M|1.0M|
> | -------- | :--------: | :--------: | :--------: | :--------: | :--------: | :--------: |
> |Vanilla LIF|886|1774|2860|3417|3697|3962|
> |PT-LIF (one-step proxy update)|886|2084|3547|4005.|4516|4744|
>
> Table C2: Average return during different training steps in the HalfCheetah-v4 environment.
> |Method|0M|0.2M|0.4M|0.6M|0.8M|1.0M|
> | -------- | :--------: | :--------: | :--------: | :--------: | :--------: | :--------: |
> |Vanilla LIF|-20|6943|8034|8559|8938|9324|
> |PT-LIF (one-step proxy update)|-20|6375|7727|8405|8951|9242|
>
> Table C3: Average return during different training steps in the Hopper-v4 environment.
> |Method|0M|0.2M|0.4M|0.6M|0.8M|1.0M|
> | -------- | :--------: | :--------: | :--------: | :--------: | :--------: | :--------: |
> |Vanilla LIF|25|1410|2594|2793|3095|3058|
> |PT-LIF (one-step proxy update)|25|1382|2690|3031|2891|2918|
>
> Table C4: Average return during different training steps in the Walker2d-v4 environment.
> |Method|0M|0.2M|0.4M|0.6M|0.8M|1.0M|
> | -------- | :--------: | :--------: | :--------: | :--------: | :--------: | :--------: |
> |Vanilla LIF|3|442|561|1057|1355|1607|
> |PT-LIF (one-step proxy update)|3|1133|2779|3696|4105|4374|
>
> In terms of resource utilization, training the proposed PT-LIF on an RTX 3090 GPU costs 21-24\% GPU utilization and about 400Mib GPU memory. In additionm an RTX 3090 GPU can support simultaneously train 10 different agents (with different environments and different seeds).
>
> In terms of comparisons under realistic training budgets,  the proposed PT-LIF model (with one-step proxy update) incurs a training time cost of only 1.35× relative to the vanilla LIF baseline, while the PT-LIF model (with default settings) cost 2.24×.
>
> Thank you again for your valuable suggestion. We will include the additional empirical evidence in the revised manuscript to provide a more rigorous evaluation of the proposed approach. Please feel free to let us know if there are any remaining concerns.
>
> References
>
> [1] Scott Fujimoto, Herke Hoof, and David Meger. Addressing function approximation error in actor critic methods. In International conference on machine learning, pages 1587–1596. PMLR, 2018.
>
> [2] Guangzhi Tang, Neelesh Kumar, Raymond Yoo, and Konstantinos Michmizos. Deep reinforcement learning with population-coded spiking neural network for continuous control. In Conference on Robot Learning, pages 2016–2029. PMLR, 2021.
>
> [3] Duzhen Zhang, Tielin Zhang, Shuncheng Jia, and Bo Xu. Multi-sacle dynamic coding improved spiking actor network for reinforcement learning. In Proceedings of the AAAI conference on artificial intelligence, volume 36, pages 59–67, 2022.
>
> [4] Ding Chen, Peixi Peng, Tiejun Huang, and Yonghong Tian. Fully spiking actor network with intralayer connections for reinforcement learning. IEEE Transactions on Neural Networks and Learning Systems, 36(2):2881–2893, 2024a.

---

### Comment · Area_Chair_R75y · 2025-08-06

Dear reviewers,

As reviewers, you are expected to stay engaged in discussion.

-  It is not OK to stay quiet.
-  It is not OK to leave discussions till the last moment.
-  If authors have resolved your (rebuttal) questions, do tell them so.
-  If authors have not resolved your (rebuttal) questions, do tell them so too.

Please note that, to facilitate discussions, Author-Reviewer discussions were extended by 48h till Aug 8, 11.59pm AoE.

Best regards,
  NeurIPS Area Chair

---

### Note · Authors · 2025-08-13

Dear Area Chair (AC) and Reviewers:

We sincerely thank you for the time you have dedicated to reviewing our work, and for the valuable insights provided throughout the review cycle.

Our contributions, as acknowledged by the reviewers, are summarized as:
1. We highlight a previously unaddressed limitation in SNN-based continuous control RL algorithms, which is the inherent conflict between discrete SNN dynamics and continuous soft-update mechanisms in RL.
2. We propose a proxy target mechanism tailored for SNN agents. It simultaneously improves stability, boosts performance, and preserves inference energy efficiency.
3. We evaluate the proposed mechanism across multiple types of spiking neurons and environments, demonstrating that SNNs trained with our framework can achieve up to 32% higher average performance.

To the best of our knowledge, this is the first work to surpass ANN performance in continuous control using a simple LIF neuron, which is appreciated by Reviewer i6Dh and Reviewer JUY9.

During the rebuttal, our responses addressed most of the reviewers' concerns. We appreciate the follow-up comments and have responded to them in detail. The major concerns are summarized as:
1. Concerns about training cost: We show the relative training time in Table R2 in response to Reviewer JUY9 and further clarify it in our discussion with Reviewer PQzp.
2. Concerns about hardware feasibility and reproducibility: We provided the hardware statistics in our discussion with Reviewer PQzp. In addition, our code was included in the supplementary materials for reproducibility.
3. Concerns about hyperparameters: We show the effects of different hyperparameters in Table R1 in response to Reviewer i6Dh. All hyperparameter settings can be found in Appendix A.3.
4. Concerns about energy efficiency: We present the average firing rates in Table R2 in response to Reviewer PQzp, the number of SOPs in Table R4 in response to Reviewer JUY9, and compare the inference energy to ANNs on NPUs in response to Reviewer tLiq.

Furthermore, we have included additional ANN-SNN conversion baselines, explained why the proposed PT-LIF achieves the best performance, and corrected some minor issues including typos and informal writings.

We gratefully accept all these constructive suggestions. We will integrate these into the next version to further strengthen our paper.

Thank you again for your careful review and thoughtful feedback.

---

### Decision · Program_Chairs · 2025-09-17

**Decision:**

Accept (poster)

**Comment:**

The authors propose a novel method for actor-critic reinforcement learning with SNNs. They introduce a proxy target network with continuous dynamics that is updated using implicit updates of weights. This stabilizes training. The method is tested on a set of continuous control tasks.
Strengths:
- A weak point in previous SNN methods is demonstrated
- The method is novel, simple, and effective.
- Good experimental validation. The method improves over the SOTA on average

The initial concerns of the reviewers were based on
- potentially high computational costs during training
- potentially high spike rates
- limited theoretical analysis
- missing ablations and analysis of varying meta parameters
- missing key baselines.
Most concerns have been settled by the authors during the discussion phase, in particular concerning computational costs and spike rates - which seem reasonable - missing ablations and analysis of varying meta parameters, and missing key baselines. One reviewer rated with 2, but did not respond to the final comments of the authors and also not to my specific question to him. So I would weight her/his rating less strongly.

The authors also provided a theoretical results, which I find however close to trivial. I do not think that it adds much to the paper. I also find the performance improvements not overly convincing, as their method improves on average over many tasks (the mean performance), but for several tasks other methods are better and standard deviations are typically large.

Despite these weak points, the manuscript pinpoints an interesting problem and proposes an innovative solution to it. I therefore weakly recommend acceptance.